# Active & Passive Causal Inference: Introduction

## Abstract

This paper serves as a starting point for machine learning researchers, engineers and students who are interested in but not yet familiar with causal inference. We start by laying out an important set of assumptions that are collectively needed for causal identification, such as exchangeability, positivity, consistency and the absence of interference. From these assumptions, we build out a set of important causal inference techniques, which we do so by categorizing them into two buckets; active and passive approaches. We describe and discuss randomized controlled trials and bandit-based approaches from the active category. We then describe classical approaches, such as matching and inverse probability weighting, in the passive category, followed by more recent deep learning based algorithms. By finishing the paper with some of the missing aspects of causal inference from this paper, such as collider biases, we expect this paper to provide readers with a diverse set of starting points for further reading and research in causal inference and discovery.

## 1 Introduction

The human curiosity and the desire to understand how things work lead to studying causality (Kidd & Hayden, 2015; Zheng et al., 2020; Ferraro et al., 2019; Bender, 2020). Causality is about discovering the causal relationship between variables. Causality delineates an asymmetric relationship where it can only say *"A causes B"* or *"B causes A"*, while correlation expresses a symmetric relationship that measures the co-occurrence of A and B. They can extend to more than two variables. Being able to depict the causal relationship is an ideal framework for humans to explain how the system works.

An important concept in causality, that we are particularly interested in, is causal effect. It refers to the impact of a choice of action on an outcome. For example, what is the impact of receiving COVID-19 vaccine on catching COVID-19? In order to know the effect of Covid-19 vaccination, we must be able to predict the outcomes of both taking and not taking the vaccine, respectively. That is, we must know the potential outcome of taking an arbitrary action. We call this process of inferring the potential outcome given an action causal inference (CI) (Rubin, 1974).

Correlation does not imply causation, and at the same time, it is possible to have causation with no correlation. CI is tricky like that because observation is not the same as intervention. The former is about passively observing what happens, while the latter is about observing what happens when one actively intervenes in the world. Taking the CI approach allows us to distinguish between correlation and causation by clearly defining the two probabilities.

To infer causal effects we must measure intervention probabilities. Intervention probability is the probability of a particular outcome resulting from our intervention in the system by imposing a specific action. This is different from conditioning, as we actively alter the action. We often have access to conditional and joint probabilities, but not intervention probabilities directly. It has thus been a major research topic to infer the intervention probability from conditional and joint probabilities in many disciplines (Rubin, 1974; Berry & Fristedt, 1985; Heckman et al., 1997; Weitzen et al., 2004; Hernán & Robins, 2006; Breslow et al., 2009; Graepel et al., 2010; Abhijit V. & Esther, 2012; Yazdani & Boerwinkle, 2015; Bouneffouf et al., 2020).

There are two main frameworks to introducing estimating causal effect, Rubin (1974)'s potential outcome framework and Pearl (2009)'s do-calculus framework. Potential outcome framework focuses on the concept of the outcomes that would have been observed under different treatment conditions. Do-calculus revolves around a set of formal rules for reasoning about intervention in a causal model. This introductory paper diverges from conventional teaching methods in causal inference by combining both Rubin's framework of potential outcomes and Judea Pearl's framework of do-calculus. We adopt a holistic approach, drawing upon concepts from both paradigms to construct a foundational understanding of causal inference rooted in first principles. Our choice to introduce potential outcomes initially stems from its intuitive appeal, particularly when illustrating treatment effects through familiar examples from domains such as medicine or the sciences. However, as we delve deeper into the formalization of causal models, the incorporation of intervention probabilities becomes essential, necessitating a shift towards joint and conditional probability distributions. By incorporating aspects of both frameworks, we aim to present a unified perspective on causal inference that facilitates a smoother transition between the intuitive conceptualization of potential outcomes and the more formalized treatment of intervention probabilities.

A causal graph is a graphical representation of causal relationships among variables in a system. It visually depicts how variables influence each other, helping us to understand and analyze the causal structure of a phenomenon. Figure 1 shows the graph representation of causal relationships for a variety of CI methods. Depending on the data collection process and experimental setup, certain methods don't require knowing the causal graph in priority, such as RCT and difference-in-difference method, while other methods require knowing the structure of a causal graph. In this paper, we consider a problem setup in which we have a covariate $X$, an action $A$, and an outcome $Y$. The action $A$ is a variable of interest and has a causal effect on the outcome. The outcome $Y$ is a variable that is affected by the treatment and is what we often want to maximize. Covariates $X$ are all the other variables that may affect and be affected by the action $A$ and the outcome $Y$. We are particularly interested in the case where these covariates are confounders, i.e., affect the action and the outcome together. We measure how treatments affect outcomes by looking at both the average effect across groups and how each person's treatment affects them personally.

There is enormous work on various assumptions and conditions that allow us to infer causal effects (Rubin, 1974). The most fundamental assumptions are i) exchangeability, ii) positivity, iii) consistency, and iv) no interference. These assumptions must be satisfied at the time of data collection rather than at the time of causal inference. When these assumptions are met, we can then convert statistical quantities, estimated from collected data, into causal quantities, including the causal effect of the action on the outcome (Hernán et al., 2019; Musci & Stuart, 2019; Zheng & Kleinberg, 2019). One way to satisfy all of these assumptions is to collect data actively by observing the outcome after randomly assigning an action independent of the covariate. Such an approach, which is often referred to as a random controlled trial (RCT), is used in clinical trials, where patients are assigned randomly to an actual treatment or placebo (Chalmers et al., 1981; Kendall, 2003; P. M. et al., 2021). RCT is deliberately designed to prevent confounding the treatment and the outcome, so that the conditional probabilities estimated from collected data approximate the intervention probabilities as well.

Randomized data collection is not always feasible and often suffers in efficiency from running large-scale experiments. There has been an enormous amount of work from various disciplines on estimating causal effects without such randomized data collection (Rubin, 1977; 1979; Chalmers et al., 1981; Lu & Rosenbaum, 2004b). As an alternative, different approaches have been proposed, including figuring out how to work with the non-randomized dataset and finding a more efficient way to collect data than the randomized approach. In this paper, we organize these CI methods into passive and active learning categories. In the passive CI category exist methods that work *given* a dataset which was *passively* collected by the experts. In contrast, the active CI category includes methods that may actively intervene in the data collection process. RCT for instance belongs to the active CI category, as it actively collects data by randomizing the treatment assignment. There are however other methods in the same category that aim also to maximize the outcome by making a trade-off between exploration and exploitation.

The organization of this literature review paper is as follows. In §2, we introduce the definitions and metrics for estimating causal effects and discuss in depth the assumptions necessary for identification of causal effects. We then cover naive conditional mean estimator and ordinary square estimator, both of which are widely

used with randomized datasets (Rubin, 1974; Pearl, 2010). In this paper, We do not consider collider bias and we assume a stationary conditional probability distribution.

In §3, we describe RCT and move on to bandit approaches in the active CI category. While bandits are used in many practical applications, the research community has been adding more emphasis on theoretical analysis of minimizing the regret bounds of different policy algorithms (Berry & Fristedt, 1985; Langford & Zhang, 2007). We look at bandits through the lens of CI, where many of the bandit algorithms can be seen as learning the classic causal graph in Figure 2b with different exploration and exploitation rates. We examine different constrained contextual bandit problems that correspond to different causal graphs, respectively. We also compare passive CI learning methods to bandits on naive causal graphs. We furthermore review causal bandits which consider graphs with unknown confounding variables (Bareinboim et al., 2015; Lattimore et al., 2016; Sachidananda & Brunskill, 2017). In this survey, we limit our scope to bandits and do not consider causal reinforcement learning which we leave for the future.

In §4, we start with classical approaches in the passive CI category, such as matching (Rubin & Thomas, 1992; Gu & Rosenbaum, 1993), inverse probability weighting (Rosenbaum & Rubin, 1983; Rubin & Thomas, 1992; Hirano et al., 2003) and doubly robustness methods (Heejung & James M., 2005; Shardell et al., 2014; Seaman & Vansteelandt, 2018). We then discuss deep learning based CI methods (Zhihong & Manabu, 2012; Pearl, 2015; Johansson et al., 2016; Wang et al., 2016; Louizos et al., 2017a). Deep learning is particularly useful when we need to conduct causal inference on high dimensional data with a very complicated mapping from input to output, as deep neural networks can learn a compact representation of action as well as covariate that captures the intrinsic and semantic similarities underlying the data (Kingma & Welling, 2014; Danilo Jimenez & Shakir, 2014). Deep learning is applied to CI in order to infer causal effects by learning the hidden/unknown confounder representations from complicated data and causal graph relationships. Such a capability of learning a compact representation from a high-dimensional input allows it to work with challenging problems such as those involving raw medical images and complex treatments (Castro et al., 2020; jiwoong Im et al., 2021; Puli et al., 2022; van Amsterdam et al., 2022).

CI is an important topic in various disciplines, including statistics, epidemiology, economics, and social sciences, and is receiving an increasingly higher level of interest from machine learning and natural language processing due to the recent advances and interest in large-scale language models and more generally generative artificial intelligence. In this paper, we cover various CI algorithms and categorize them into active and passive CI families. The goal of this paper is to serve as a concise and readily-available resource for those who are just starting to grow their interest in causal inference.

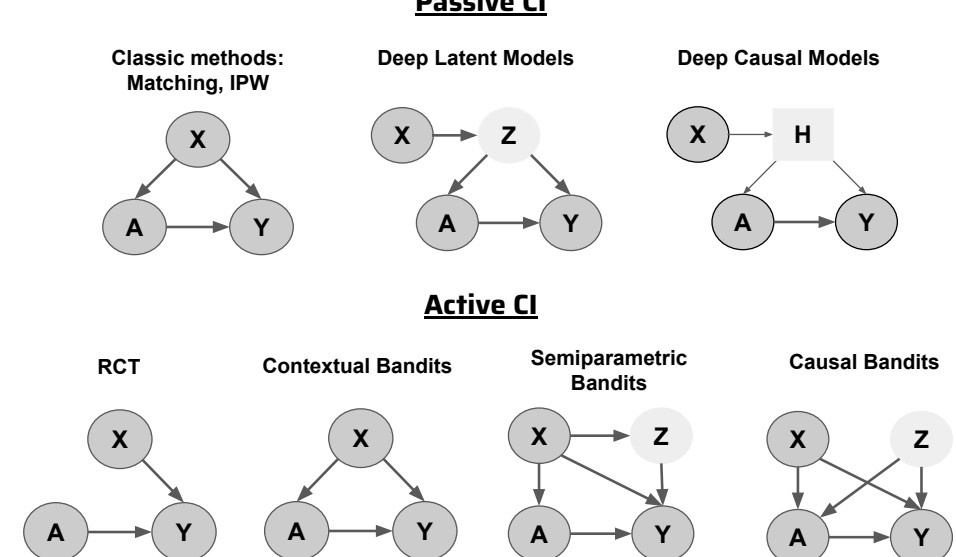

Figure 1: Examples of passive and active causal inference methods. Dark gray nodes correspond to observed variables while light gray nodes correspond to latent variables. A square node corresponds to a deterministic variable while a circle corresponds to stochastic variables.

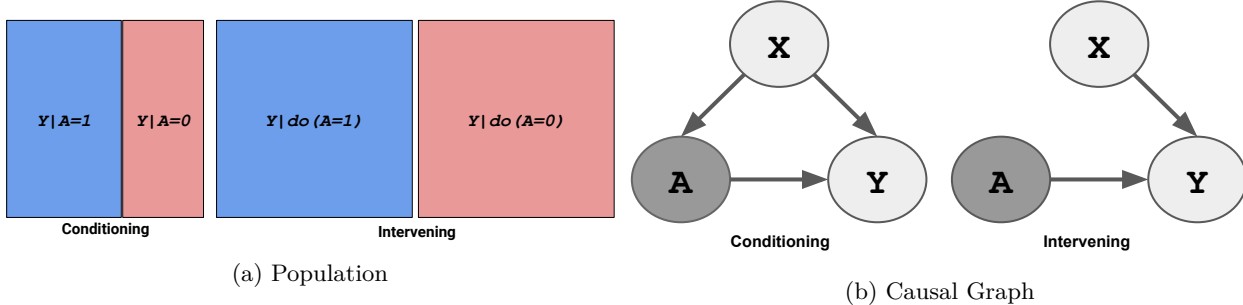

(a) Population
(b) Causal Graph

Figure 2: Condition versus intervention

## 2 Background

### 2.1 Preliminary

Let $X$, $A$ and $Y$ be *covariate*, *action*, and *outcome* variables, respectively. We define the *joint*, *conditional* and *intervention* probabilities as

$$\text{Joint: } p(Y = y, A = a, X = x) = p(X = x)p(A = a|X = x)p(Y = y|A = a, X = x),$$

$$\text{Conditional: } p(Y = y|A = a) = \frac{\sum_x p(X = x)p(A = a|X = x)p(Y = y|A = a, X = x)}{\sum_{x',y'} p(X = x')p(A = a|X = x')p(Y = y'|A = a, X = x')}, \text{and}$$

$$\text{Intervention: } p(Y = y|do(A = a)) = \sum_x p(X = x)p(Y = y|A = a, X = x),$$

respectively. We marginalize the covariate variable to obtain the conditional and intervention probabilities. We observe that the conditional probability is often different from the intervention probability, $p(Y|A = a) \neq p(Y|do(A = a))$. The conditional probability takes into account the prevalence of a particular action

$a$ in the population and checks how often a particular outcome $y$ is associated with it. On the other hand, the intervention probability does not consider the prevalence of the action $a$ and only considers what the outcome would be had the action been forced to be $a$. $p(Y|A = a, x)$ distribution tells us the effect of $A$ on $Y$ given $X = x$, since $A = a$ is assigned directly. $p(Y|do(A = a))$ is a marginal distribution of $p(Y|A = a, x)$ over $X$ and $A = a$ was directly assigned. As soon as one incorporates $p(A = a|X)$ or $p(A = a)$, it is not an intervention but a joint. See Figure 2a for a graphical illustration. This can be understood as removing all the incoming edges to the action variable, i.e. $X \to A$, when we intervene on $A$, as in Figure 2b. The intervention probability formula above reflects it by ignoring the probability of a particular action. The intervention probability $p(Y|do(A))$ describes the causal effect of action $A$. The corresponding causal graph $A \Rightarrow Y$ is a graphical representation used in causal inference to depict the causal relationships between variables in a system (see Figure 2b right).

Consider the following example that shows how intervention and conditional probabilities are sometimes different and sometimes the same, given two variables.

**Example 1.** *let $C$ indicate whether coffee is hot or cold and $T$ be the thermometer reading that measures the temperature of the coffee. Based on our everyday observation, $T$ and $C$ are highly correlated. For instance, $p(C = hot|T = 70°)$ and $p(T = 70°|C = hot)$ are both high. However, it is clear that forcing the thermostat's reading to be high does not cause the temperature of coffee to go up, that is, $p(C = hot|do(T = 70°))$ is low despite high $p(C = hot|T = 70°)$. On the other hand, boiling coffee would indeed cause the thermostat's reading to go up, that is, both $p(T = 70°|C = hot)$ and $p(T = 70°|do(C = hot))$ are high.*

Another example, demonstrating the discrepancy between the intervention and conditional probabilities, is *Simpson's paradox* in which different assumptions about outcome, treatment, and confounder variables lead to different conclusions on causation:

**Example 2** (Simpson's paradox illustration (Carlson, 2019))**.** *You are studying sex bias in graduate school admission. According to the data, men were more likely to be admitted to graduate school than women were, where 40% of male applicants and 25% of female applicants were admitted. In other words, there was a strong association between being a man and being admitted. You however found that such association varies across different sub-populations. For example, 80% and 46% of men and women were admitted to natural science respectively, and 20% and 4% of women and men were admitted to social science respectively. It turns out that the social science department has a much lower acceptance rate than the natural science department, while women were more likely to apply to social science and men were more likely to apply to natural science departments. In summary, we can derive different conclusions about sex and admission rate association by either combining or separating sub-populations.*

*We can notice how adding the department to the covariate variable or not changes the result. If the department is not part of the covariate, then the conditional and intervention probability coincide with each other, since there is no confounding via the department choice. Otherwise, these two probabilities deviate from each other due to $p(S = s|X = x)$. This demonstrates that CI is inherently dependent upon our modelling assumption.*

A *potential outcome* is an outcome given a covariate under a potential action (Rubin, 1974; 2005). Assuming a binary outcome variable $Y \in \{0, 1\}$ and a discrete action variable $A \in \mathcal{A}$, the potential outcome (Rubin, 1974) is defined as

$$Y_X(a) = Y|do(A = a).$$

For instance, there are two potential outcomes, $Y_X(1)$ and $Y_X(0)$, for a binary action $A \in \{0, 1\}$. It is often impossible to compute these potential outcomes given a fixed covariate directly due to the fundamental problem of causal inference, that is, we cannot perform two distinct actions for a given covariate $X$ simultaneously and observe their outcomes (Saito & Yasui, 2019). We can only observe the outcome of one action that has been performed, to which we refer as the *factual outcome*, but cannot observe that of the other action, to which we refer as the *counterfactual outcome*. We instead consider the potential outcome averaged over the entire population, represented by the covariate distribution $p(X)$. We call it the *expected potential outcome* $\mathbb{E}_{Y,X}[Y_X(A = a)]$, as opposed to the *conditional potential outcome*, where we marginalize out the covariate $X$.

The goal of causal inference is to estimate the potential outcomes that each individual would have experienced under different treatment conditions, as well as the average or expected outcomes across the population, based on observed data, $(y_i|do(A = a_i), x_i, a_i) \overset{\text{iid}}{\sim} p$, where $p$ is the data distribution and $a_i \in \mathcal{A}$ is the action performed for the $i$-th covariate $x_i$.

We often hear about the *treatment effect* of an experimental drug or a new surgical procedure in the medical literature. Treatment effect measures whether and how much the treatment caused the difference in the outcome (or in the potential outcome). Often treatment effect is used for binary actions in medical research. Unless confusing, we will interchangeably use the treatment effect and causal effect throughout this paper. In general, the treatment effect is defined as the difference between two potential outcomes $Y_X(1) - Y_X(0)$. The *average treatment effect* (ATE) is then the difference between the potential outcomes averaged over the covariate distribution (Rubin, 1974; Imbens, 2004):

$$ATE := \mathbb{E}_{X,Y}[Y_X(1) - Y_X(0)] = \mathbb{E}_{X,Y}[Y_X(1)] - \mathbb{E}_{X,Y}[Y_X(0)]. \tag{1}$$

We may be interested in the average treatment effect over a subpopulation, defined by a subset $X' \subseteq X$ of covariates. We then compute the *conditional average treatment effect* (CATE). CATE is defined as averaging the treatment effect for an individual patient characterized by $X'$ (Radcliffe, 2007; Athey et al., 2015):

$$CATE(x') := \mathbb{E}_{Y,X\setminus X'}[Y_X(1) - Y_X(0)|X' = x'] = \mathbb{E}_{Y,X\setminus X'}[Y_X(1)|X' = x'] - \mathbb{E}_{Y,X\setminus X'}[Y_X(0)|X' = x'], \tag{2}$$

where $X\setminus X'$ is the remainder of the covariate over which we compute the expectation.

There are a few alternatives to the ATE. The first one is an *expected precision in the estimation of heterogeneous effect* (PEHE (Imbens, 2004)):

$$PEHE := \mathbb{E}_{X,Y}[(Y_X(1) - Y_X(0))^2].$$

Another alternative is the *population average treatment effect* (PATT) for the treated, i.e. $a = 1$ (Rubin, 1977; Heckman & Robb, 1985):

$$PATT(a) := \mathbb{E}_{X,Y|A=a}[Y_X(1) - Y_X(0)],$$

which is a helpful quantity when a particular, treated sub-population is more relevant in the context of narrowly targeted experiments. All of these metrics have their own places. For instance, it is more common to see PEHE in medical research, while PATT can be used to study the effect on the treated group programs (e.g. individuals disadvantaged in the labour market (Heckman & Robb, 1985)).

## 2.2 Assumptions for Causal Inference

Unfortunately, we cannot simply average the factual outcomes for each action to estimate the average treatment effect, because this corresponds to measuring the expected conditional outcome which is a biased estimate of the expected potential outcome. This bias is usually due to confounders. For instance, socioeconomic status can be a confounder in the study of the treatment effect of a medication. Socioeconomic status often affects both patients' access to medication and their general health, which makes it a confounder between the treatment and health outcome. In this case, the expected conditional outcome estimate is biased, because those patients who receive the medication are also likely to receive better healthcare, resulting in a better outcome. We must isolate the effect of medications on the health outcome by separating out the effect of having better access to healthcare due to patients' higher socioeconomic status, in order to properly estimate the expected treatment effect. In this section, we review the assumptions required for us to obtain an unbiased estimator for the average potential outcome.

The main strategy for estimating the (average) potential outcome is to compute causal quantities, such as intervention probabilities, from statistical quantities that are readily estimated from a set of samples, i.e., a dataset. In this context, we can say that a causal quantity is *identifiable* if we can compute it from statistical quantities. In doing so, there are a number of assumptions that must be satisfied.

**Positivity/Overlap.** The first step in estimating the potential outcome is to estimate the conditional probabilities from data. In particular, we need to compute

$$p(Y|X, A) = \frac{p(Y, A|X)}{p(A|X)}$$

for $X$ with $p(X) > 0$. This implies that $p(A = a|X)$ for the action $a$, of which we are interested in computing the potential outcome, must be positive. We call it the *positivity*. Positivity is a necessary condition for computing ATE as we need the $p(A|X = x) > 0$ in the denominator for data $x$. The *overlap* assumption is similar to the positivity assumption but applies to the covariate. It requires that the distributions $p(X|A = 0)$ and $p(X|A = 1)$ have common support. Partial overlap occurs when you are missing on particular action for a certain area of covariate space. For example, we can have treated units of certain patient groups but no control units, or vice versa.

**Ignorability/Exchangeability.** Even if we can estimate the statistical quantities, such as the conditional probability $p(Y|X, A)$, we need an additional set of assumptions in order to turn them into causal quantities. The first such assumption is *exchangeability* which states that the potential outcome $\hat{Y}(a)$ must be preserved even if the choice of an action to each covariate configuration $p(A|X)$ changes. That is, the causal effect of $A$ on $Y$ does not depend on how we assign an action $A$ to each sample $X$. This is also called *ignorability*, as this is equivalent to ignoring the associated covariate when assigning an action to a sample, i.e., $A \perp\!\!\!\perp X$. This enables us to turn the conditional probabilities into intervention probabilities, eventually allowing us to estimate the potential outcome, which we describe in more detail later.

The exchangeability is a strict condition that may not be easily met in practice. This can be due to the existence of confounding variables, selection bias, or due to time-dependency between action selections (violation of Markov Assumption). We can relax this by assuming *conditional exchangeability*. As we did for defining the CATE above, we partition the covariate into $X$ and $X'$ and condition the latter on a particular value, i.e., $X' = x'$. If the exchangeability is satisfied conditioned on $X' = x'$, we say that conditional exchangeability was satisfied. This however implies that we are only able to estimate the potential outcome given a particular configuration of $X'$. To meet the ignorability assumption, we need to achieve unconfoundedness, which refers to having an absence of confounding variables in a causal relationship. This involves careful design, data collection, and measurements to control potential confounders and isolate the impact. These two assumptions together allow us to turn the statistical quantities into causal quantities, just like when (unconditional) exchangeability alone was satisfied.

**Consistency and Markov assumption.** Data for causal inference is often collected in a series of batches rather than at once in parallel. Each $t$-th batch has an associated potential outcome $\hat{Y}(a_t)$ for action $a$. If the potential outcome changes over these batches, we cannot concatenate all these batches and use them as one dataset to estimate the potential outcome. That is, we must ensure that $\hat{Y}(a_t) = \hat{Y}(a_{t'})$ for all $t, t'$ where $a_t = a_{t'}$. This condition is called *consistency*. Furthermore, we must ensure that the past batches do not affect the future batches in terms of the potential outcome, i.e., $\hat{Y}(a_1, \ldots, a_t) = \hat{Y}(a_t)$, as this would effectively increase the action space dramatically and make it impossible to satisfy positivity. We call this condition a *Markov* assumption.

In practice, the potential outcomes are estimated from the data, which is a statistical estimation. Hence, all assumptions are required to turn causal estimand into statistical estimand. We show step-by-step how each assumption is used to compute ATE:

$$
\begin{aligned}
ATE &= \mathbb{E}_X[\mathbb{E}_{X'}[Y(1) - Y(0)|X']] \\
&= \mathbb{E}_X[\mathbb{E}_{X'}[Y(1)|X'] - \mathbb{E}_{X'}[Y(0)|X']] && \text{(Conditional exchangeability)} \\
&= \mathbb{E}_X[\mathbb{E}_{X'}[Y(1)|A = 1, X']] - \mathbb{E}_X[\mathbb{E}_{X'}[Y(0)|A = 0, X']] && \text{(Ignorability)} \\
&= \mathbb{E}_X[\mathbb{E}[Y|A = 1, X]] - \mathbb{E}_X[\mathbb{E}[Y|A = 0, X]] && \text{(Consistency)} \\
&= \mathbb{E}[Y|A = 1] - \mathbb{E}[Y|A = 0].
\end{aligned}
$$

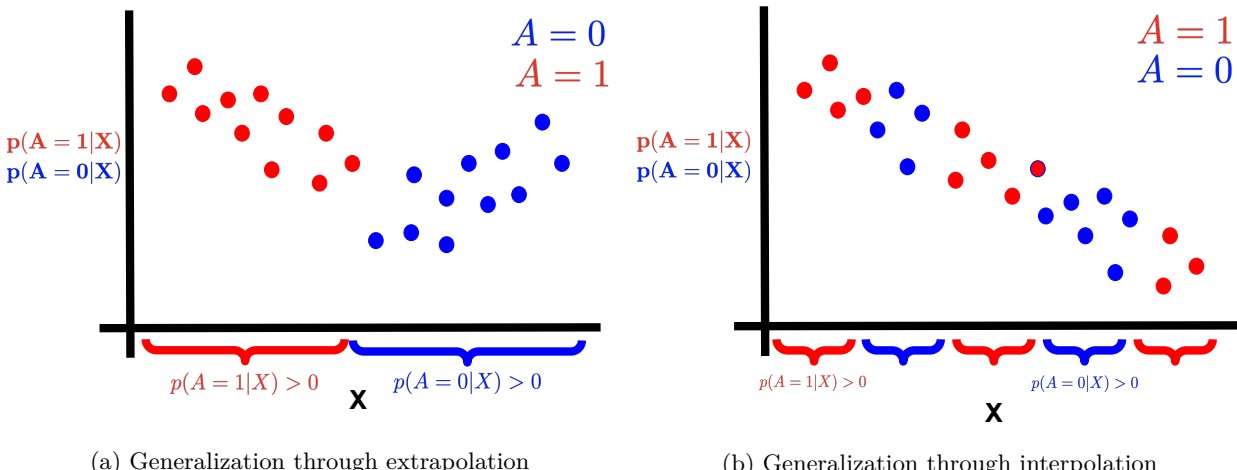

(a) Generalization through extrapolation       (b) Generalization through interpolation

Figure 3: Generalization of the propensity score in two scenarios where the positivity assumption is violated. (a) requires extrapolation and (b) requires interpolation for generalizing to unseen counterfactual examples respectively.

## 2.3 Discussion on the assumptions in practice

These assumptions above, or some of their combinations, enable us to derive causal quantities from statistical quantities. It is thus important to carefully consider these assumptions when faced with causal inference and how they may be violated, as most of these are often impossible to verify in practice. We discuss a few important points regarding these assumptions in practice.

**Unconfoundedness & Conditional Exchangeability.** We face the challenge of verifying whether the potential outcome remains the same across all possible confounder configurations because we cannot enumerate all possible confounders. In practice, we use conditional exchangeability because we often condition each individual data point (e.g., covariates per patient). However, the conditional exchangeability assumption is impossible to test and verify in practice. Estimating the potential outcome given a particular configuration of $X'$ means removing all the existing confounders for $X'$, however, we cannot know every possible confounder out there. In practice, we condition $X' = x'$ to be an individual data point (e.g., $x'$ being a patient) and often conditional exchangeability is taken for granted for that data point.

**Unconfoundedness vs. Overlap.** In order to estimate ATE, one must assume both unconfoundedness and positivity. It is however difficult to satisfy both of them in practice, because there is a natural trade-off between them. We are likely to satisfy the unconfoundedness by adding more features to the covariate, which in turn increases the dimensionality of data. This in turn increases the chance of violating the overlap assumption due to the curse of dimensionality. Similarly, we can satisfy the overlap assumption by choosing only the minimum number of features as a covariate, but we may unintentionally create unobserved confounders along the way.

**Positivity via generalization.** Positivity is hard to satisfy in a strict sense, as we must have as many data points as there are actions for each and every possible $x$ with $p(x)$. This is largely impossible when $x$ is continuous, and even when $x$ is discrete, it is difficult if the support of $p(x)$, i.e., $\{x \in \mathcal{X} | p(x) > 0\}$ is large. Instead, we can fit a parametric or non-parametric model to model $p(A = a|X)$ that can generalize to an unseen combination of $(X, A)$. In such a case, the success of generalization depends on the support of data points we have access to. Figure 3 presents a depiction of two cases where the fitted model must respectively extrapolate and interpolate. In the latter case of interpolation, even without positivity, we may be successful at correctly inferring the causal effect, but in the former case, it would be much more challenging. This suggests we must be careful at relying on generalization to overcome the issue of positivity (or lack thereof.)

---

**Algorithm 1** Active CI protocol

---

$A$ actions, $T$ rounds (both known); potential outcome $Y(a)$ for each action $a$ (unknown *a priori*).
**while** each round $t \in [T]$ **do**
    Observe a covariate $x_t$
    Pick an action according to policy, $a_t \sim \pi(x)$.
    Outcome observed $y_t \in [0, 1]$ is sampled given $X = x_t, A = a_t$.
    **if** action $a$ sampled from $\pi(x) = p(A|X = x)$ **then**
        Update the potential outcome $\mathbb{E}_{p(X)}[Y_X(a)] \leftarrow t^{-1} \sum_{t'=1}^{t} \frac{\mathbb{I}(a_{t'})}{p(a|x)} y_t$
    **else**
        Update the potential outcome $\mathbb{E}_{p(X)}[Y_X(a)] \leftarrow t^{-1} \sum_{t'=1}^{t} \frac{\mathbb{I}(a_{t'})}{p(a)} y_t$
    **end if**
    [Optional] Update the policy $\pi$.
**end while**

---

# 3 Active Causal Inference Learning

Because counterfactual information is rarely available together with factual information in observed data (Graepel et al., 2010; Li et al., 2010; Chapelle et al., 2015), one approach to causal inference (CI) is to design an online algorithm that actively explores and acquires data. We introduce and describe an active CI framework that combines causal inference and data collection, inspired by works of literature on contextual bandit(Slivkins, 2019; Bouneffouf et al., 2020). In this framework, an algorithm estimates the expected potential outcome $Y_X(A)$ of each action and collects further data for underexplored counterfactual actions. A general active CI protocol is presented in Algorithm 1. We denote $(x_t, a_t, y_t)$ for observed covariate, action, and outcome at time $t$, and denote $Y_X(A)$ for potential outcome. We update the estimated expected potential outcome $\mathbb{E}_X[Y_X(A)]$ based on a new observed data point $(x_t, a_t, y_t)$. Although active CI algorithms are inspired by contextual bandit, there is a major difference between these two; that is, CI focuses much more on estimating the expected potential outcomes, while bandit algorithms on finding an optimal policy function that maximizes the expected outcome values.[1]

Most of the time CI researchers and practitioners face the challenge of being limited to data exploration not by our choice, but by other factors that make it inaccessible to A/B testing from real-world constraints. Active CI learning benefits from the policy function $\pi(x)$ when it can compromise between exploration and exploitation for understanding the causal effect and optimizing the decision in real-world applications respectively. In this section, we review active CI literature by first examining RCT and then expanding it to contextual bandit methods.

## 3.1 Randomized Controlled Trial

Randomized controlled trial (RCT) is the most well-known and widely practiced CI method. It is *de facto* standard in e.g. clinical trials, where participants are randomly divided into treatment and control groups, and the outcomes of these two groups are used to compute the average treatment effect (ATE). RCT falls under the active CI method group since it collects data proactively by randomly assigning an action to each data point (Rubin, 1974). Although it is typical to choose a uniform distribution over an action set $\pi = \mathcal{U}[\mathcal{A}]$ as a policy function in RCT, we can in principle pick any random distribution as long as it is independent of the covariate $\pi = \mathcal{P}[\mathcal{A}]$, as shown in Algorithm 2.[2]

A set of points $\{(x_t, a_t, y_t)\}_{t=0}^{T}$ collected by RCT automatically satisfy the exchangeability assumption. Because actions are selected independently of covariate $X$ as shown in Figure 2b, the potential outcome $Y_X[A]$, estimated from this data, must be preserved even if action $P(A|X)$ changes. With these we can show that the conditional and intervention probabilities coincide with each other, allowing us to perform causal

---

[1]We use *outcome* and *reward* interchangeably.
[2]Existing literature often conflates having an equal chance of sampling each action with independence, but this is not true.

---
**Algorithm 2** Randomized Controlled Trial

---
Observes a covariate $x_t$.
Pick an action according to a policy, $a_t \sim \pi = \mathcal{P}[\mathcal{A}]$.
Observe the outcome $y_t \in [0, 1] \sim Y_X(A)|X = x_t, A = a_t$.
Estimate $\mathbb{E}_{p(X)}[\hat{Y}_X(a)]$ for all $a$.

---

inference from data collected by RCT:

$$
\begin{aligned}
p(Y = y | do(A = a)) &= \sum_x p(Y = y | A = a, X = x) p(x) \\
&= \sum_x \frac{p(Y = y | A = a, X = x) p(A = a | X = x) p(x)}{p(A = a | X = x)} \\
&= \sum_x \frac{p(Y = y, A = a, X = x)}{p(A = a | X = x)} \\
&= \sum_x \frac{p(Y = y, A = a, X = x)}{p(A = a)} \quad (\text{since } A \perp\!\!\!\perp X) \\
&= \sum_x p(Y = y, X = x | A = a) \\
&= p(Y = y | A = a).
\end{aligned}
\tag{3}
$$

Let us consider Simpson's paradox (Ex. 2 in §2.3). Although there was an overall strong association between being a man and being more likely to be admitted, we found different associations when we considered different sub-populations due to the uneven sex distribution among the applicants and the acceptance rates, across departments. With RCT using the uniform action distribution, we would end up with an even number of each sex independent of the department choice, i.e., $p(A = a | S = \text{man}) = p(A = a | S = \text{woman})$. This would allow us to verify whether men are more likely to get admitted to graduate school without being confounded by the department's choice.

It is often tedious and difficult to plan and design an RCT due to many biases that are difficult to mitigate. One such example is a control bias which arises when the control group behaves differently or is treated differently from the treatment group. For instance, participants in the control group may be more likely to drop out of the study than those in the treatment group, due to the lack of progress they perceive themselves. In order to avoid a control bias, one must carefully consider eligibility criteria, selection of study groups, baseline differences in the available population, variability in indications in covariates, and also the management of intermediate outcomes (Simon, 2001; Jane-wit et al., 2010). There are techniques that help you assess the quality of RCT studies with an emphasis on measuring control bias in the planning and implementation of experiment designs (Chalmers et al., 1981; 1989; Stewart & Parmar, 1996; Moreira & Susser, 2002; Olivo et al., 2008).

RCT is widely used in clinical trials in medicine (Concato et al., 2000; Rothwell, 2005; Green & Glasgow, 2006; Frieden, 2017), policy research in public health (Sibbald & Roland, 1998; García & Wantchekon, 2010; Deaton & Cartwright, 2016; Choudhry, 2017), econometrics (Heckman & Robb, 1985; LaLonde, 1986; Abhijit V. & Esther, 2012) and advertisements in marketing (Graepel et al., 2010; Chapelle et al., 2015; Gordon et al., 2019). In these real-life applications, we often cannot afford to blindly assign actions but to determine the action based on the covariate $X$, due to ethical, legal and economical reasons. Because it is challenging to apply RCT in practice (Saturni et al., 2014), some studies explore combining RCT with observational study data (Rubin, 1974; Concato et al., 2000; Hannan, 2008). For example, one can use inverse probability weighting or matching techniques to de-bias the potential outcome estimation, as we will discuss in §4. We thus review active CI methods with a covariate-dependent data collection policy in the rest of this section.

### 3.2 Causal inference with contextual bandits

It is often impractical to use RCT in real-life settings due to (but not limited to) the following three limitations (Rubin, 1974; Olivo et al., 2008; Schafer & Kang, 2009; Saturni et al., 2014). First, a sample size must be large enough to detect a meaningful difference between the outcomes of actions, due to the high variance of RCT. Second, complete randomization or complete independence from the covariate is often neither feasible nor ethical in practice. Lastly, complete randomization often goes against the real-world objective of maximizing the outcome which is different from correctly inferring the causal effect. For example, doctors should not randomly assign different treatments to patients in order to test their causal effects on the patients, because this could end up harming many patients. Instead, a doctor makes the best decision for each patient given their expertise (i.e. their own policy), and on the fly adjusts their policy online in order to maximize the outcome of each patient:

$$\arg\max_{\pi_e} \mathbb{E}_{x \sim p(X)} \mathbb{E}_{a \sim \pi_{e(x)}} \left[ Y_X(a) \right].$$

RCT on the other hand does not maximize the outcome at all, which makes it less desirable to use in many real-world scenarios.

In this section, we review different ways of performing both tasks together, that is, finding an optimal policy and estimating causal effects. In particular, we examine various ways to intervene and actively collect data under the framework of contextual bandits.[3] There are two primary approaches to applying contextual bandit algorithms to CI. The first approach actively gathers interventional data and then estimates ATE from this actively collected randomized dataset.[4] The second approach uses a contextual bandit method to learn a causal model utilizing all the collected data points including both randomized and non-randomized actions.

#### 3.2.1 Estimating ATE from interventional data collected with a bandit method

The general idea is to keep track of data points for which intervention, i.e. randomization, happened. It is no different from RCT except that random intervention happens only occasionally based on your choice of bandit algorithm. For the purpose of illustration, we use the $\epsilon$-greedy strategy together with the expert's policy as an example in Algorithm 3. The epsilon greedy algorithm is an easy way to add exploration to the basic greedy algorithm; we greedily choose an action based on the estimated highest outcome values but once in a while randomly select an action independently of the covariate with the probability $\epsilon$. We use a *randomized set* to refer to a collection of these randomized actions together with associated outcomes and covariates. We then estimate the average treatment effect (ATE) directly using the randomized dataset only (see Appendix A.1). The efficiency in estimating the causal effect is determined solely by how often we randomize action, i.e., the probability $\epsilon$. In contrast to RCT, as we select actions based on the covariate-aware policy occasionally with the probability $1 - \epsilon$, we fulfill both outcome maximization and ATE estimation, although the efficiency in ATE estimation is typically worse than that of RCT.

There are other, more sophisticated methods such as high-confidence elimination and upper-confidence bound (UCB) algorithms Auer et al. ("2002"); Slivkins (2019).[5] Unlike the $\epsilon$-greedy strategy, these approaches choose when to randomize action based on a learned policy so far. A *high-confidence elimination method* alternates between two actions, $a$ and $a'$ until the confidence bounds of the two actions' potential outcomes do not overlap, where the confidence bounds are defined as

$$UBC_t(a) = \mathbb{E}\left[Y(a)\right] + r_t(a)$$
$$LBC_t(a) = \mathbb{E}\left[Y(a)\right] - r_t(a)$$

with the confidence radius $r_t(a) = \sqrt{2\log(T)/n_t(a)}$[6] and the number $n_t(a)$ of rounds with the action $a$ Slivkins (2019).

---

[3] See Appendix A.3.1 for the basic description of contextual bandits.

[4] A randomized dataset refers to a dataset consisting of tuples collected using actions chosen independently of covariates.

[5] See Appendix A.3.1 for more details.

[6] There are ways to estimate a tighter radius based on different assumptions according to the previous research Slivkins (2019).

---

**Algorithm 3** $\epsilon$-greedy protocol

---

$A$ actions, $T$ rounds (both known); potential outcome $Y(a)$ for each action $a$ (unknown).
**while** In each round $t \in [T]$ **do**
    Toss a coin with the exploration probability $\epsilon_t$.
    **if** explore **then**
        explore: choose an action $a_t \sim \mathcal{U}[a]$
    **else**
        Observe a covariate $x_t$
        Pick an action according to the expert $a_t \sim \pi_e(x)$.
    **end if**
    Observed the outcome $y_t \sim Y|X = x_t, A = a_t \in [0,1]$.
    Store $y_t$ in set $\mathcal{D}$ if explore
    Update the expected potential outcome $\mathbb{E}_{p(X)}[Y_X(a)] \leftarrow |\mathcal{D}|^{-1} \sum_d^{\mathcal{D}} \mathbb{I}[A = a] y_d$ for all $a$.
**end while**

---

Once the lower-confidence bound of the potential outcome of one action is greater than the upper-confidence bound of that of the other action, i.e., $UCB(a) < LCB(a')$, the former action $a'$ is selected indefinitely from there on because the abandoned action cannot be the best action in terms of maximizing the outcome value. The rationale behind this method is to make sure to explore until we are confident about the expected potential outcome of each action. In short, the high-confidence elimination method fully explores until the best action is determined with a high level of confidence, after which it converts to exploiting the discovered best action only. The first phase of exploration is thus similar to performing RCT. This approach is interesting because as soon as the lower bound of $a'$ and the upper bound of $a$ separate, we automatically get some level of assurance about the potential outcome estimates.

The *UCB algorithm* on the other hand picks an action that maximizes $UCB_t(a)$ at every round $t$. This choice automatically balances exploration and exploitation. $a'$ is selected over another action $a$ if $UCB_t(a') > UCB_t(a)$, which can happen for one of two reasons; the uncertainty is high (exploration) and the actual reward is high (exploitation). For the purpose of estimating ATE, we only use the randomized (explored) data points where the confidence radius $r_t(a)$ was large. There are other methods such as Thompson-sampling causal forest Dimakopoulou et al. (2017) and random-forest bandit Féraud et al. (2016), which share a similar flavour with the UCB algorithm.

### 3.2.2 Learning a causal graph using a bandit algorithm

So far in this section, we have discussed how to estimate ATE directly by gathering an interventional dataset using bandit methods. While such an approach allows us to measure the expected potential outcomes and ATE, we cannot infer conditional potential outcomes. One of the CI goals is to measure the causal effect using ATE but a bigger and more ambitious goal is to answer counterfactual questions. The latter is only possible if we can infer individual potential outcomes. Here, we discuss how to learn an underlying graphical model using a contextual bandit method in order to infer individual potential outcomes.

At each trial $t$, we approximate the potential outcome $\hat{Y}_X(a_t)$ using a parametrized model $g_\theta(x_t, a_t)$ that computes the outcome based on the context $x_t$ and action $a_t$,

$$\hat{Y}_X(a_t) = g_\theta(x_t, a_t) + \epsilon_t, \tag{4}$$

where $\epsilon_t \sim \mathcal{N}(0, \sigma^2)$ is noise which comes from unknown confounders at time $t$. Although data was collected by the generic probabilistic graph that includes a policy $\pi$ in Figure 4b, $g_\theta$ learns a causal graph in Figure 4a, because the parameters $\theta$ are estimated from the intervention dataset alone. This allows us to use $g_\theta$ to infer the potential outcome of a counterfactual action and/or of unseen covariates.

Contextual bandits can handle causal graphs that are more complicated. For instance, in Fig. 4c, there is an extra variable $Z = f(X)$ that mediates the effect of the covariate $X$ on the outcome $Y$ but does not affect

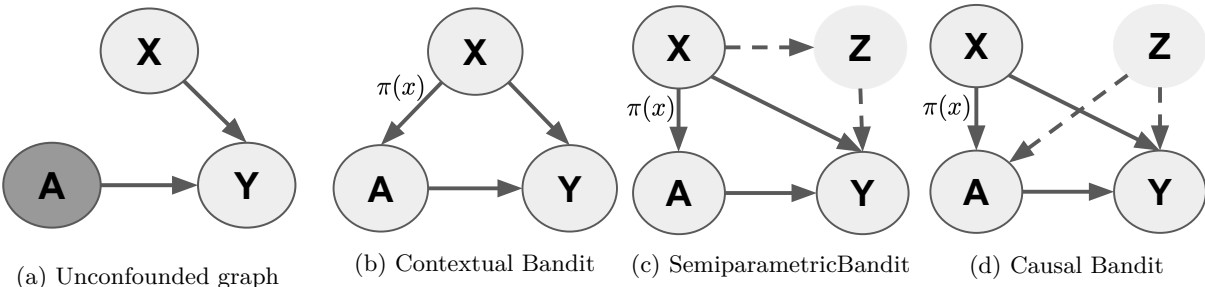

Figure 4: Bandit methods for Active CI

the action $A$. In this case, the outcome function is in the form of

$$\hat{Y}_X(a_t) = g_\theta(x_t, a_t) + f_\phi(x_t) + \epsilon_t,$$

where $g_\theta(x_t, a_t)$ takes into account the confounder $X$ as well as policy function $\pi(X)$ and maps them to an output $Y$. $f_\phi(X)$ learns to isolate the effect of the covariates that affect the outcome independent of the action. $\epsilon_t$ is the noise. Having $f_\phi(X)$ to learn covariates that are not confounders can reduce the complexity of learning for $g_\theta(x_t, a_t)$.

When both $g_\theta$ and $f_\phi$ are non-parametric, it is often computationally intractable to tackle this problem. In order to avoid these issues of computational tractability and undesirable regret bound, *semiparametric contextual bandits* consider parametric policies (Krishnamurthy et al., 2018; Greenewald et al., 2017; Peng et al., 2019) and learn a linear bandit model using regression oracles to estimate potential outcome (Swaminathan et al., 2017)

$$\hat{Y}_X(a_t) = w^\top x_t + f_\phi(x_t) + \epsilon_t,$$

where $w^\top$ is a parameter vector and $\epsilon_t$ is noise. They have a nice property where the regret bound is the same as the regular contextual bandit's regret. The action-independent features $f_\phi(x_t)$ get cancelled out in the regret formulation together with the noise term $\epsilon_t$, because they are independent of the action choice.

### 3.2.3 Correcting a bias from unobserved confounders

It is unrealistic to observe all confounders in practice, nor for us to verify whether all confounders have been observed. Any confounding factor not included in the covariate leads to difficulties in accurately estimating the causal effect. Here, we consider a way to detect and correct such a bias that arises from having unobserved confounders (see Figure 4d).

Consider a policy function $\pi^*(x')$ that is optimized for maximizing $\arg\max_a CATE(x', a)$:[7]

$$\pi^*(x') = \arg\max_a CATE(x', a) = \arg\max_a \mathbb{E}_{Y, X \setminus X'} \left[ Y_X(a) - \max_{\bar{a}} Y_X(\bar{a}) | X' = x' \right].$$

This $\pi^*(x')$ is estimated without considering unobserved confounders like in Figure 4b, and yet, with the realization of unobserved confounders shown in Figure 4d, we now correct the estimation. Let $\bar{a}$ be the counteraction to $\pi^*(x')$ that returns a larger CATE than the originally selected action $\pi^*(x')$. Suppose we found $x'$ where $CATE(x', \pi^*(x')) < CATE(x', \bar{a})$ where $\bar{a} \neq \pi^*(x')$. Since the policy function $\pi^*$ is optimal, this can only happen if there are unobserved confounders. This illustrates that our learned policy might not be the best in such a situation. We then need to search for a new policy function $\pi'$ that can handle this situation better.

---

[7]Unlike $CATE(x)$ with a binary action in Equation 2, we define $CATE(x, a)$ for a set of more than two actions.

Because our counteraction could not be better than the action $\pi^*(x')$ without unobserved confounders, we compare the potential outcomes of the action and the counteraction $\bar{a}$;

$$\mathbb{E}_{Y,X\setminus X'}\left[Y_X(\pi^*(x'))|X'=x'\right] \text{ and} \tag{5}$$

$$\mathbb{E}_{Y,X\setminus X'}\left[Y_X(\bar{a})|A=\pi^*(x')|X'=x'\right], \tag{6}$$

respectively. In order to compute the outcome of the counteraction, we actively intervene with the counteraction and collect a new sample $(x', y', \bar{a}, a)$. We also count the frequency of the expected potential outcome of counteraction being higher than action $\pi^*(x')$,

$$f(\pi^*) = \sum_i^N \mathbb{I}\left[\mathbb{E}_{Y,X\setminus X'}\left[Y_X(\pi^*(x_i'))|X'=x_i'\right] > \mathbb{E}_{Y,X\setminus X'}\left[Y_X(\bar{a})|A=\pi^*(x_i'), X'=x_i'\right]\right],$$

where $\frac{f(\pi^*)}{N}$ tells us how often $\pi^*$ is wrong. Our new policy function $\pi'(x')$ samples a new action according to the Bernoulli distribution with probability $\frac{f(\pi^*)}{N}$. This strategy copes with the fact that our policy is biased and incorrect with the probability $\frac{f(\pi^*)}{N}$ due to unobserved confounders.

Bareinboim et al. (2015) show that while this method requires collecting enough data for $f$ to converge, we can optimize a new policy function faster by weighting the samples from the Beta distribution $\mathcal{B}(f, (1-f))$ by the bias as shown in Algorithm 4. The bias from the unobserved confounder is defined as one minus the absolute treatment effect (Bareinboim et al., 2015),

$$\text{bias} = 1 - |\mathbb{E}_{Y,X\setminus X'}[Y_X(\bar{a})|A=\pi^*(x'), X'=x'] - \mathbb{E}_{Y,X\setminus X'}[Y_X(\pi^*(x'))|X'=x']|. \tag{7}$$

Eq. 7 quantifies how effective applying action $\pi^*(x')$ over the counteraction $\bar{a}$ is. If this quantity is large, the re-weight of the probability of choosing action $\pi^*(x')$ by the bias remains high. Otherwise, the re-weight of the probability of choosing counteraction $\bar{a}$ by bias remains high. Overall this encourages faster convergence of re-estimating the potential outcome (see the last three lines of Algorithm 4). This specific algorithm is called causal Thompson sampling.

---

**Algorithm 4** Causal Thompson Sampling (TS$^\text{C}$) (Bareinboim et al., 2015)

---

Let $a = \pi^*(x')$ be intuitive action.
Let $\bar{a}$ be counteraction to $a$.
Let $\mathbb{E}_{Y,X\setminus X'}[Y_X A = a | A = a, X' = x']$ be the expected payout for intuitive action
Let $\mathbb{E}_{Y,X\setminus X'}[Y_X A = \bar{a} | A = a, X' = x']$ be the expected payout for counter-intuitive action

**while** $t = 1 \cdots T$ **do**
    Let $w = [1, 1]$
    Sample $a \sim \text{intuition}(x_t, t)$

    // Estimate the potential outcomes and bias
    Compute $bias = 1 - |\mathbb{E}[Y_X(\bar{a})|A = a, X' = x'] - \mathbb{E}[Y_X(a)|X' = x']|$ (Equation 7)
    **if** $\mathbb{E}[Y_X(\bar{a})|A = a, X' = x'] > \mathbb{E}[Y_X(a)|X' = x']$ **then**
        Set $w[a] = bias$
    **else**
        Set $w[\bar{a}] = bias$
    **end if**

    // Choose a new policy $\pi'(x')$ based on new weighting
    $\beta_1 = \mathcal{B}(f(a), (1 - f(a)))$
    $\beta_2 = \mathcal{B}(f(\bar{a}), (1 - f)(\bar{a}))$
    Set $\dot{\pi}(x') \leftarrow \max(\beta_1 \cdot w[a], \beta_2 \cdot w[\bar{a}])$
    Set $\dot{y} = \text{simulate}(\dot{\pi}(x'))$
    Update $\mathbb{E}[Y_X(\dot{\pi}(x'))|A = a, X' = x']$ with $(x', \dot{y}, \dot{\pi}(x'), a)$
**end while**

---

## 4 Passive Causal Inference

Unlike in active causal inference (CI), in passive CI, we must calculate ATE given a non-randomized dataset which was gathered in advance with the actions chosen based on covariates (Rubin, 1974; Holland, 1986). As discussed earlier in §2.3, we assume that the dataset satisfies positivity, although we discuss how to relax it with deep learning later in this section. In this section, we present and examine passive approaches to CI, which are gaining more popularity. Most of them are primarily grounded in one of the following three general approaches: matching, inverse probability weighting and doubly-robust methods. We first review these basic approaches and introduce some of the more recent approaches.

### 4.1 A Naive Estimator

Before we begin, we start with a naive estimator $\mu_A(X)$ of the outcome variable $Y$. Let $\mu_A : \mathcal{X} \to \mathcal{Y}$ be the generic estimator that predicts the outcome. For example, this estimator can be simple empirical averaging:

$$\mu_a(x) = \frac{\sum_{i=1}^{N} y_i \mathbb{I}[a_i = a, x_i = x]}{\sum_{i'=1}^{N} \mathbb{I}[x_{i'} = x]},$$

where $D = \{(x_i, a_i, y_i)\}_{i=1}^{N}$ is a dataset that consists of $N$ data points, and $\mathbb{I}$ is an indicator function. This estimator looks at the average outcome of the action $a$ given a particular covariate $x$. Another example would be to have a parametrized estimator, $\mu_A(X; \theta)$ where $\theta$ is a parameter.

Such a naive estimator $\mu_A(X)$, which often maximizes the log-likelihood, is a biased estimator of the potential outcome $Y$, due to the discrepancy between the conditional and invention probabilities. In the subsequent sections, we introduce and discuss modifications either to the estimator or to the dataset, that allows us to obtain an unbiased estimate of the potential outcome.

## 4.2 Matching

ATE estimation is hard when working with a real-life dataset due to confounding. A common approach is to construct a randomized dataset, that is free of confounding, from a non-randomized one. The matching method achieves this by pairing each treated instance with a control instance and ignoring any unmatched instance. This process balances the treatment and controlled group (Scotina & Gutman, 2019).

Ideally, we should have both factual and counterfactual outcomes for every data point ($\langle (x_i, y_i(1), y_i(0) \rangle$ for all $i$). This is often impossible due to the fundamental problem of CI, where the problem arises from the very fact that we can only observe the outcome of a single action given a particular covariate. We instead aim for approximate one-to-one matching where we pair treated data with controlled data that are similar enough. That is, we pair two instances if $\{\langle (x_i, 1, y_i(1)), (x_j, 0, y_j(0)) \rangle | D(x_i, x_j) < \epsilon \text{ for } i \neq j\}$ and for some small $\epsilon$, where $D(\cdot, \cdot)$ is a problem-specific distance metric. We then compute ATE using our new balanced dataset. Matching methods remove confoundedness by creating comparable groups based on observed covariates. It selects individuals from different treatment groups but has similar characteristics or covariate distributions. This ensures that the treatment and control groups are balanced concerning the potential confounders. The advantages and disadvantages of such a matching method lie in the bias-variance tradeoff. The advantage of the matching method is that it reduces the confounding bias, but it increases the variance because we removed (potentially many) unmatched data points.

There are many standard metrics of *closeness* that are widely used. One such metric is Mahalanobis distance:

$$D_{ij} = (x_i - x_j)^\top \Sigma^{-1} (x_i - x_j),$$

where $\Sigma$ is the covariance metric. Many alternative metrics have been proposed over the past decades, such as those relying on a coarsened data space or other feature spaces (Cochran & Rubin, 1973; Rubin, 1979; Rosenbaum & Rubin, 1983; Rubin & Thomas, 1992; Rubin & Stuart, 2006; Stuart, 2010; Iacus et al., 2012; Zubizarreta, 2012; Zhao, 2004; Resa & Zubizarreta, 2016). The choice of a metric must be determined for each problem separately.

Matching methods have evolved from a greedy algorithm based (heuristic-based search) to optimal/full matching (Kim & Steiner, 2016). The greedy method ends up with a sub-optimal solution since the ordering of pairing matters (Weitzen et al., 2004). Moving away from greedy matching, one can use an optimal non-bipartite matching algorithm that runs in polynomial time (Lu & Rosenbaum, 2004a; Dehejia & Wahba, 1999). Such an algorithm generates a series of matched sets that consist of at least one treated individual and at least one control individual. It is optimal in terms of minimizing the average distance between each treated individual and each controlled individual within each matched set (Hansen, 2004). There are other approaches such as weighting methods where one adjusts the importance of distance between data points (Heckman et al., 1997; Hirano et al., 2003; Imbens, 2004). These methods can be helpful when the data samples are unevenly spread out over the domain.

There however remain challenges with matching. First, it is difficult to have exact matches for all data points with a non-randomized or imbalanced dataset (the dataset is unevenly distributed w.r.t actions). We thus end up eliminating a significant number of unpaired data points after matching between treated and controlled groups, resulting in the loss of information and statistical power. There are however some algorithms that enable many-to-one matching, with the simplest being the $K$-nearest neighbour method (Karp, 1972; Schafer & Kang, 2009; Zubizarreta, 2012). Second, matching works well when the dimensionality of the covariate is low, but it easily fails when the dimensionality is high due to the curse of dimensionality (Gu & Rosenbaum, 1993). It can be helpful to use deep learning to obtain a more concise and dense representation, similar to what we will discuss in §4.5.

## 4.3 Inverse Probability Weighting

Inverse probability weighting (IPW) removes the effect of confounders by weighting the potential outcome of each action by its inverse probability weight (Rosenbaum & Rubin, 1983; Robins et al., 1994; Hirano et al.,

2003):

$$\mathbb{E}\left[\frac{Y\mathbb{I}[A=a]}{p(A|X)}\right] = \mathbb{E}_{p(X)}\left[\mathbb{E}\left[\frac{Y(a)\mathbb{I}[A=a]}{p(A|X)}\Big|X\right]\right] = \mathbb{E}\left[\frac{\mathbb{E}\left[Y(a)\Big|X\right]\mathbb{E}\left[\mathbb{I}[A=a|X]\right]}{p(A|X)}\right] = \mathbb{E}\left[Y(a)\right]. \quad (8)$$

Equation 8 illustrates that we can take the subset of data that corresponds to a particular action $a$ as long as we can divide by the so-called propensity score in order to compute $\mathbb{E}[Y(A)]$. $p(A = a|X = x)$ is known as the *propensity score* $e(x)$.[8]

The propensity score theorem (Rosenbaum & Rubin, 1983) tells us that if ignorability is satisfied given $X$, then ignoreability conditioned on $e(X)$ is also satisfied (Imbens & Rubin, 2015):

$$(Y(1), Y(0)) \perp\!\!\!\perp A|X \implies (Y(1), Y(0)) \perp\!\!\!\perp A|e(X).$$

This predicate tells us that the potential outcomes are independent of $A$ given confounder $X$, and this is due to the blocking back-door criterion conditioning on confounder $X$ (Pearl, 2009). Similarly, the potential outcomes are independent of $A$ given the propensity score $e(x)$, and the propensity score has the same effect as removing the blocking back-door path in a causal graph by conditioning on the edge between $X$ and $A$. This illustrates that the 1-dimensional score function, that is the propensity score, is enough to compress the high-dimensional confounder $X$.

While simple finite-sample averaging $\widehat{ATE}_{AGG}$ is a biased estimator of ATE,

$$\widehat{ATE}_{AGG} = \frac{1}{N}\sum_i^n\left[\frac{\mathbb{I}[A_i=1]y_i(1)}{\eta(x_i)}\right] - \frac{1}{N}\sum_i^n\left[\frac{\mathbb{I}[A_i=0]y_i(0)}{1-\eta(x_i)}\right],$$

$\widehat{ATE}_{IPW}$ is an unbiased estimator of ATE,

$$\widehat{ATE}_{IPW} = \frac{1}{N}\sum_i^n\left[\frac{\mathbb{I}[A_i=1]y_i(1)}{e(x_i)}\right] - \frac{1}{N}\sum_i^n\left[\frac{\mathbb{I}[A_i=0]y_i(0)}{1-e(x_i)}\right],$$

where $\eta(x) = \frac{N_{x1}}{N}$ and $N_{x1}$ is the number of treated samples. Note that $\widehat{ATE}_{AGG}$ uses a naive estimator from §4.1 to estimate ATE. $\widehat{ATE}_{IPW}$ is known as *an oracle IPW estimator* since the propensity score $e(\cdot)$ is known. The estimation error, defined as $\sqrt{N}|ATE - \widehat{ATE}_{IPW}|$, follows a Normal distribution with zero mean and the variance of

$$\text{VAR}_{IPW} = \text{Var}[ATE_{AGG}(X)] + \mathbb{E}_p(x)\left[\frac{\sigma^2(X)}{e(X)(1-e(X))} + \frac{c(X)^2}{e(X)(1-e(X))}\right], \quad (9)$$

where $c(X)$ is a function that satisfies

$$Y(0) = c(X) - (1 - e(X))ATE(X) + \epsilon(X)$$
$$Y(1) = c(X) + e(X)ATE(X) + \epsilon(X),$$

where $\epsilon(X)$ is a Gaussian noise with variance of $\sigma^2(x)$. $\widehat{ATE}_{AGG}$ can be seen as a version of the IPW estimator with an imperfect propensity score being $\hat{e}(x) = \frac{N_{x1}}{N}$. $\text{VAR}_{IPW}$ demonstrates that even with using

---

[8]Additionally, propensity scores can be used for matching methods where one compares two covariates with similar propensity values $D(\hat{e}(x_i), \hat{e}(e_j))$ for $i \neq j$ and some distance metric

$$D_{ij} = |e(x_i) - e(x_j)|,$$

where $e(x_i)$ and $e(x_j)$ are the propensity scores for the data point $x_i$ and $x_j$, respectively. The propensity score is a popular method as it summarizes the entire covariate into a single scalar and has been shown to be effective in theory (Rosenbaum & Rubin, 1983; Rubin & Thomas, 1992; Rubin & Stuart, 2006; Zubizarreta, 2012; Diamond & Sekhon, 2013; Resa & Zubizarreta, 2016; Abadie & Imbens, 2016) and in practice (Jalan & Ravallion, 2001; Dehejia & Wahba, 2002; Monahan et al., 2011; Amusa, 2018).

the true propensity score $e(x)$, $\widehat{ATE}_{IPW}$ has a worse asymptotic variance than $\widehat{ATE}_{AGG}$ which can be thought of as inverse probability reweighting with an incorrect propensity score $\hat{e}(x)$. This is an example of the bias-variance trade-off.

A plethora of methods have been proposed since then, that are unbiased and exhibit lower variance than the oracle IPW above by replacing the propensity score with other weighting schemes $\hat{e}'$. For example, one can reduce the variance of an IPW estimator by normalizing the weights (Hirano et al., 2003):

$$\widehat{ATE}_{SW} = \frac{\sum_i^n \mathbb{I}[A_i = 1]y_i(1)w(x_i)}{\sum_i^n \mathbb{I}[A_i = 1]w_1(x_i)} - \frac{\sum_i^n \mathbb{I}[A_i = 0]y_i(0)w_0(x_i)}{\sum_i^n \mathbb{I}[A_i = 0]w_0(x_i)},$$

where $w_1 = \frac{1}{e(x_i)}$ and $w_0 = \frac{1}{1-e(x_i)}$. This leads to a lower variance in the estimate, and these weights are thus called *stabilized weights* (Robins et al., 2000). According to Hirano et al. (2003), $\widehat{ATE}_{SW}$ outperforms $\widehat{ATE}_{IPW}$ in terms of asymptotic convergence rate (Hirano et al., 2003). Lunceford & Davidian (2004) review various versions of the IPW estimator and suggest a way to take into account the uncertainty in estimating the propensity score using a closed-form sandwich estimator (M-estimator (Stefanski & Boos, 2002)) (Lunceford & Davidian, 2004).

### 4.4 Doubly Robust Methods

It is often hard to obtain the propensity score in advance nor guarantee that our propensity estimate $\hat{e}(x)$ is accurate. Furthermore, even with the oracle propensity score, we have just shown that the IPW has a high variance. On the other hand, the naive estimator (without IPW) from §4.1 is unbiased only when the actions in the dataset were sampled independently of the associated covariates. In other words, it is often not enough to rely on either of these approaches on their own to perform causal inference (Belloni et al., 2011). We can do better by combining IPW with the naive estimator $\mu_A(X)$, to which we refer as a doubly robust estimator. In doubly robust estimation, a bias from one method is addressed by the other method, and vice versa (Robins et al., 1994; Lunceford & Davidian, 2004; Kang & Schafer, 2007; Chernozhukov et al., 2017). Such a method is both consistent and unbiased, as long as at least one of the IPW and the naive estimator is consistent and unbiased. The doubly robust estimator for ATE in the case of two actions is then

$$ATE_{DR} = \mathbb{E}_{Y,X}\left[\mu_1(X) - \mu_0(X)\right] + \mathbb{E}_{Y,X,A}\left[A\frac{(Y - \mu_1(X))}{e(X)} - (1-A)\frac{(Y - \mu_0(X))}{1-e(X)}\right] \quad (10)$$

$$= \mathbb{E}_{Y,X,A}\left[A\frac{(Y - \mu_1(X))}{e(X)} + \mu_1(X)\right] - \mathbb{E}_{Y,X,A}\left[(1-A)\frac{(Y - \mu_0(X))}{1-e(X)} + \mu_0(X)\right]. \quad (11)$$

If the estimated potential outcomes $\mu_A(X)$ are correct, we do not need to worry about propensity score estimation, since $Y - \hat{\mu}_1(X)$ and $Y - \hat{\mu}_0(X)$ will be zero in Equation 11. $\widehat{ATE}_{DR}$ hence reduces to approximating $\mathbb{E}\left[\hat{\mu}_1(X) - \hat{\mu}_0(X)\right]$. In contrast, it is okay for our estimated potential outcome to be wrong if propensity score estimation is consistent. If the propensity score is correctly estimated, then $\mathbb{E}_{Y,X,A}\left[\frac{AY}{\hat{e}(X)}\right]$ and $\mathbb{E}_{Y,X,A}\left[\frac{(1-A)Y}{(1-\hat{e}(X))}\right]$ will be weighted correctly as well, and we recover the IPW estimator (Robins et al., 1994; Robins & Rotnitzky, 1995; SCHARFSTEIN et al., 1999). Consequently, such a doubly robust method is consistent (Hahn, 1998; Heejung & James M., 2005; Shardell et al., 2014; Farrell, 2015).

Re-arranging terms and expressing in terms of Monte Carlo estimation of $ATE_{DR}$, we get

$$\widehat{ATE}_{DR} \approx \frac{1}{N}\sum_i^N\left[\frac{a_iy_i}{\hat{e}(x_i)} - \frac{a_i - \hat{e}(x_i)}{\hat{e}(x_i)}\hat{\mu}_1(x_i)\right] - \frac{1}{N}\sum_i^N\left[\frac{(1-a_i)y_i}{1-\hat{e}(x_i)} - \frac{\hat{e}(x_i) - a_i}{1-\hat{e}(x_i)}\hat{\mu}_0(x_i)\right]. \quad (12)$$

The empirical estimate converges the true ATE, $\hat{ATE}_{DR} \to ATE$, with an asymptotic variance of

$$\mathrm{Var}[ATE_{DR}(X)] = \mathrm{Var}[ATE_{AGG}(X)] + \mathbb{E}\left[\frac{\sigma_1^2(X)}{e(X)}\right] + \mathbb{E}\left[\frac{\sigma_0^2(X)}{1-e(X)}\right],$$

where $\sigma_T(X) = \text{Var}[Y_i(T)|X]$. Despite its greater variance, the doubly robust method often exhibits greater efficiency and robustness to model misspecification.

Multiple studies have shown that doubly robust methods for ATE estimation with missing data perform better than their non-doubly robust baselines and theoretically have shown to converge faster to the true ATE than individual methods (Robins et al., 1994; Mayer et al., 2020). A naive doubly-robust estimator is asymptotically optimal among non-parametrized estimators, meaning the semiparametric variances are bounded and asymptotically convergent if either the propensity score or the estimator $\mu_A(X)$ is correct (Robins & Rotnitzky, 1995; Kang & Schafer, 2007). Subsequently, other estimators with bounded asymptotic variances have been proposed even when IPW exhibits a high variance (Robins et al., 2007; Tan, 2010; Waernbaum & Pazzagli, 2017).

### 4.5  Causal inference with representation learning

While matching, IPW, and doubly robust methods have their own merits for estimating potential outcomes, it may be necessary to utilize a powerful model that can express highly complex functions given high dimensional data with a limited amount of data. For instance, working with non-linear high-dimensional data such as X-ray images may require a non-linear parametric model to transform data from its original space to a better representation that facilitates CI. The goal is to extract a causal representation from high dimensional data (covariate and action) and more accurately predict potential outcomes from the extracted causal representation. In this section, we review methods that extend the previous CI approaches in this way by (deep) representation learning (Schölkopf et al., 2021; Wang & Jordan, 2021).

Representation learning involves automatically learning features or representations of raw data. Rather than manual feature engineering, representation learning enables models to learn to extract high-level features from raw, high-dimensional data such as images, audio, and texts, themselves. Here, the potential outcome estimator $u_A(X; \phi)$ is a deep neural network with parameter $\phi$ with $m$ hidden layers $H = \{\mathbf{h}^{(i)}\}_1^m$. Each hidden representation at $i$-th layer ($i < k$) is a function of all the previous layers, $\mathbf{h}_i(x) = f_i(\mathbf{h}_{i-1}; \phi)$ with $\mathbf{h}_0 = x$. Each hidden representation at $j \geq k$, $\mathbf{h}_j(x) = f_j(\mathbf{h}_{j-1}, a; \phi)$, depends also on the action.

The model is trained to minimize the factual loss $l_{\text{Factual}}$ which is typically the negative log-likelihood of the dataset with inverse probability weighting. $u_A(X; \phi)$ learns to predict the potential outcome however only well on observed covariate-action pairs due to the challenges in generalization. We thus need to add a regularization term to the loss function in order to encourage the model to generalize better to (unseen) counterfactual actions:

$$\mathcal{L} = \mathbb{E}_{Y,X,A}[wl_{\text{Factual}}(X, Y, A; u_A(X; \phi)) + \lambda \mathcal{R}(H)]. \tag{13}$$

The loss function is weighted by the inverse probability weight $w = \frac{A}{2e(X)} + \frac{1-A}{2(1-e(X))}$ (see §4.3). Regularization often imposes certain properties on the hidden layers, which is why we refer to it as $\mathcal{R}(H)$, with the regularization coefficient $\lambda$ (Johansson et al., 2016; Uri et al., 2017; Yao et al., 2020; Wu & Fukumizu, 2021). Regularization reduces the hypothesis space in a way that encourages $\mu_A(X, \phi)$ to capture a causal relationship rather than a spurious relationship between the action and outcome. Deep representation learning based CI is vast and fast-growing (Nabi & Shpitser, 2017; Yoon et al., 2018; Veitch et al., 2020; Zhang et al., 2020; Wang & Jordan, 2021; Zhang et al., 2021). We characterize a majority of these approaches as minimizing a combination of a factual regression loss and a regularizer, as in Equation 13, and we discuss four representative ones in this section.

Among these methods, we separately discuss counterfactual smoothing regularization and deep latent variable models (see Figure 5) with a focus on how both of these approaches lead to better generalization of ATE.

### 4.5.1  Counterfactual Smoothing Regularization

The objective in counterfactual smoothing is to train a model to generalize to a counterfactual potential outcome even if it only saw factual data during training (see Figure 5a). Although the model $u_A(X, \phi)$ is trained to estimate the potential outcome using the inverse probability weighted factual loss function, it may still underperform for an unseeded data or a data paired with unseen action (see Figure 3). In such a

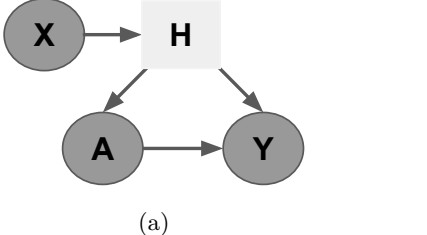 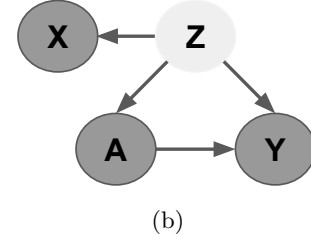

(a)             (b)

Figure 5: Causal Graph - (a) Deterministic Representation: $H$ corresponds to a hidden layer representation that is a deterministic variable, $X$ is a covariate, $A$ is an action, and $Y$ is an outcome variable. (b) Stochastic Representation: $Z$ is a latent variable, $X$ is a noise version of confounder $Z$, $A$ is an action, and $Y$ is an outcome variable.

case, the variance of the potential outcome estimate over counterfactual actions is high, implying that an individual model's prediction may be inaccurate.

Counterfactual smoothing reduces the variance of the predicted potential outcome whose variance can be controlled by ensuring that the learned representations of treated and controlled data points to follow (largely) indistinguishable distributions in the feature space, respectively (Johansson et al., 2016; Shalit et al., 2017; Johansson et al., 2018; Yao et al., 2018). The divergence between factual and counterfactual hidden representations' distributions influences the maximum variance of the ATE since covariate and an action propagate through deep neural networks, assuming that the deep neural network has a finite Lipschitz constant. We can thus reduce the variance by optimizing the factual loss function while, for instance, minimizing the integral probability metric (IPM) as regularization. The overall objective function is

$$\mathcal{L}_{CFR} = \mathbb{E}_{Y,X,A}[wl_{\mathrm{Factual}}(X, Y, A; u_A(X; \phi))] + \lambda \mathrm{IPM}_{\mathcal{F}}(\hat{p}(\mathbf{h}_i(X)|A = 1), \hat{p}(\mathbf{h}_i(X)|A = 0)), \quad (14)$$

where $\mathbf{h}_i(X)$ is a hidden representation from the $i$-th layer of $u_A$ with $i < k$. $\mathrm{IPM}_{\mathcal{F}}$ is the integral probability metric with a class $\mathcal{F}$ of real-valued bounded measurable functions. The hidden representation $\mathbf{h}_i(X)$ does not depend on action $A$, although the data point $x_i$ was collected together with some action $a$. $\hat{p}(\mathbf{h}_i(X)|A = 1)$ and $\hat{p}(\mathbf{h}_i(X)|A = 0)$ are thus the empirical probability distributions over the representations of treated and controlled groups, respectively. The 1-Lipschitz function and universal reproducing Hilbert kernel space, such as 1-Wasserstein distance (Cuturi & Doucet, 2013) and MMD (Gretton et al., 2012), is the most commonly used class of functions for IPMs (Shalit et al., 2017; Johansson et al., 2018).

Suppose $\mathrm{IPM}_{\mathcal{F}}(\hat{p}(\mathbf{h}_i(X)|A = 1), \hat{p}(\mathbf{h}_i(X))|A = 0) = 0$. Then, we would not be able to tell whether the hidden representation $\mathbf{h}_i$ is likely conditioned on action or counterfactual action. The generalization error of ITE is bounded by the CFR objective and it has been empirically shown to have lower ATE for the out of sample data. However, this does not theoretically guarantee that generalization error will be always lower. We explain how the approach upper bounds the PEHE estimation in Appendix A.4.2.

Zeng et al. (2020) extend this approach to use a doubly robust estimator instead of IPW (see §4.4) and simultaneously minimize the Jensen-Shannon divergence between the treated and controlled group instead of IPM. Hassanpour & Greiner (2019) view ITE estimation problem from a domain adaption perspective where factual data is assumed to come from the source and the counteraction data from a different target distribution. They use importance sampling to re-weight the loss function for each factual data to make it look like it was sampled from the target distribution instead of the source distribution (Hassanpour & Greiner, 2019). Instead of globally balancing the treatment and controlled posterior distributions, Yao et al. (2018) propose a local similarity preserved individual treatment effect (SITE) estimation method based on deep representation learning (Yao et al., 2018). Their method preserves the local similarity and balances between the factual and counterfactual distributions over the set of actions, simultaneously. Furthermore, there is a line of work where they separately extract the representations of confounders and non-confounders and re-weight the confounder representation only (Kuang et al., 2017; Wu et al., 2020).

Domain invariance, or equivalently the full overlap between the factual and counterfactual distributions, can be an overly restrictive criterion, as it may remove information from input variables (covariates and sometimes

action) that may be necessary for accurately estimating the treatment effect. (Stojanov et al., 2021). Yao et al. (2020) demonstrate why it is not ideal to use a distributional divergence to balance the treated and controlled representations (Yao et al., 2020). Instead, they propose to minimize the counterfactual variance and make the hidden representation invertible by adding a reconstruction loss function, which is, they claim, enough to have sufficient overlap in the factual and counterfactual supports.

*Deep kernel learning for individual treatment effect* (DKLITE) is a representative passive CI method that uses variance reduction (Yao et al., 2020). Unlike the methods above, this algorithm manipulates the action-dependent hidden representation. More formally, it uses kernel regression to estimate the potential outcome $\hat{y}_i = W_a \mathbf{h}_{m-1}(x_i) + \epsilon_{i,a}$ on top of the final layer of deep neural network $\mathbf{h}_{m-1}(x)$, where $W_a$ is the parameter for action $a$ and $\epsilon_{i,a}$ is an action-dependent noise variable. The posterior distribution is $\mathcal{N}(m_a \mathbf{h}_{m-1}(x), \sigma^2(x; \mathcal{X}, \Theta_a))$, where $\sigma^2(x; \mathcal{X}, \Theta_a) = \mathbf{h}_{m-1}^T K_a^{-1} \mathbf{h}_{m-1}$ is the variance. DKLITE objective function minimizes both the negative log-likelihood and posterior variance given the counterfactual actions. Although DKLITE uses kernel regression to derive the posterior distribution, it is not necessary to use kernel regression.

The objective function for training a causal inference model with posterior variance reduction can be written as

$$\mathcal{L}_{VR} = w\mathcal{L}_{\text{Factual}}(X, Y, A; u_A(X; \phi)) + \lambda \mathbb{E}_{p(X,A)}[g(\text{VAR}[h_\theta(X, 1-A)])], \qquad (15)$$

where $\text{VAR}[h(X, 1-A)]$ is the posterior variance given a data point $X = x$ and a counterfactual action. The function $g : \mathbb{R}^D \to \mathbb{R}$ aggregates the covariance of high dimensional representations into a single scalar. For example, $g(\cdot)$ can be a sum of the element-wise variances of the hidden representation. By reducing the variance in the representations $h(X, 1-A)$, we also reduce the variance in the ATE.

### 4.5.2 Deep Latent Variables CI Models

In deep latent variable models for causal inference, we assume a particular data-generating process described with a probabilistic graphical model that contains latent (or hidden) stochastic variables (Louizos et al., 2017a; Rakesh et al., 2018; Vowels et al., 2020; Wu & Fukumizu, 2021; jiwoong Im et al., 2021; Kumor et al., 2021; Lu et al., 2022). The inclusion of such latent variables enables us to model the potential outcome with a much richer distribution. Without latent variables, it is challenging to build a function approximator, such as a deep neural network, that captures a multimodal distribution of the potential outcome, regardless of how complex a form such a function takes.

Causal effects are however not identifiable in general when there are latent variables. Identifiability requires that the model parameter can be uniquely estimated from the data. However, since the latent variables do not directly measure the unobserved variable but infer from the observed variables, it introduces ambiguity that leads to violates the assumption of unconfoundedness. In order to overcome this issue, one has to make two assumptions; (1) $X$ is a proxy variable that is a noisy version of a hidden confounder, and (2) this unknown confounder can be modelled by these latent variables (see Figure 5b). While these extra assumptions are a major drawback of deep latent variable-based CI (Louizos et al., 2017b; Kocaoglu et al., 2017), we nevertheless review this literature as they are increasingly more widely used in practice (Pearl et al., 2016; Wu & Fukumizu, 2021; Trifunov et al., 2020; Kumor et al., 2021; Rissanen & Marttinen, 2021).

With these assumptions, we can consider the latent variable $Z$ a hidden, i.e. unobserved, confounder, to which we have access via its noisy realization $X$. We can recover the joint distribution $p(Z, X, Y, A)$ from observational data $(X, Y, A)$. We can then compute $p(Y|X = x, do(A = 1))$ and $p(Y|X = x, do(A = 0))$, which allows us to compute ITE, by

$$p(Y|X = x, do(A = a)) = \int_z p(Y|X, do(A = a), Z)p(Z|X = x, do(A = a))dZ$$

$$= \int_z p(Y|X = x, do(A = a), Z)p(Z|X = x, do(A = a))dZ$$

$$= \int_z p(Y|X = x, A = a, Z)p(Z|X = x)dZ.$$

The third equality follows from the rule of *do*-calculus. Therefore, we can estimate $p(Y|X = x, do(A = a))$ as long as we can approximate $p(Y|A = a, Z)$ and $p(X|Z)$.

The causal effect variational autoencoder (CEVAE) is a particular type of variational inference framework which allows us to estimate $p(Y|A, Z)$ and $p(Z|X)$ using deep neural networks (Kingma & Welling, 2014; Louizos et al., 2017a). With this VAE, the true posterior distribution is defined as $p_\theta(Z|X, Y, A) \propto p_\theta(Y, A|Z)p_\theta(X|Z)p(Z)$, where both $p_\theta(Y, A|Z)$ and $p_\theta(X|Z)$ are modelled using a deep neural network parametrized by $\theta$ and $p(Z)$ is a prior distribution and is not parameterized by $\theta$. We approximate the posterior distribution with a variational posterior $q_\phi(Z|X, Y, A)$ which is modelled by a deep neural network with a variational parameter $\phi$. We infer the hidden confounder $Z$ from the observation $(X, Y, A)$ using this variational posterior neural network.

We estimate $p(Y|A, Z)$ and $p(Z|X)$ directly by training both generative and inference networks on observational data. Training is done to maximize the following variational lower bound with respect to the parameters $\theta$ and $\phi$:

$$\mathcal{L}_{\text{CEVAE}} = \mathbb{E}_{p_{data}(X,Y,A)} \left[ \mathbb{E}_{q_\phi(Z|X,Y,A)} \left[ \log p_\theta(X, Y, A|Z) \right] - \mathbb{KL} \left[ q_\phi(Z|X, Y, A) \| p(Z) \right] \right].$$

The first term is the reconstruction of observable variables from the inferred confounder $Z$, and the the second term is a regularizer which enforces the approximate posterior to be close to the prior and maximizes the entropy of the posterior distribution over the confounder $Z$. We jointly update both generative and inference network parameters by using backpropagation and the re-parameterization trick (Danilo Jimenez & Shakir, 2014; Kingma & Welling, 2014).

Similar to CEVAE, linked causal variational autoencoder (LCVA) treats the latent attributes directly as confounders with the assumption that these confounders affect both the treatment and the outcome of units (Rakesh et al., 2018). The main difference is that the authors want to measure the causal effect when there exists *a spillover effect*[9] between pairs of two covariates through the confounders. Another variant is the Causal Effect by using Variational Information Bottleneck (CEVIB) (Lu et al., 2022). Just like any other variational latent model, it learns to fit the model to observation data and learns the confounders that affect treatments and outcomes using variational information bottleneck (Alemi et al., 2016). CEVIB does this in a way that allows the model to forget some latent variables that are not confounders and learn to extract only the confounding information from covariate. Deep entire space cross networks for individual treatment effect estimation (DESCN) attempt to learn the latent confounders ("the hidden treatment effect") through a cross-network in a multi-task learning manner (Zhong et al., 2022). It reduces treatment biases, that favour one treatment over another, by learning from multiple tasks and overcoming sample imbalance.

In CEVAE, the conditional distribution $p_\theta(X, Y, A|Z)$ is learned from data sampled from $p(A|Z)$, $p(X|Z)$ and $p(Y|A, Z)$, where $Z \sim p(Z)$. This conditional distribution, which is used for computing treatment effect, is however used with an action $A$ sampled from $p(A)$ rather than $p(A|Z)$ in the inference time. This discrepancy is known as covariate shift or more generally distribution shift and is detrimental to generalization in deep learning (Shimodaira, 2000; Sergey & Christian, 2015; Jeong & Namkoong, 2020; Louizos et al., 2017b), which in turn results in a degradation in the quality of ATE estimation.

Replacing the observational distribution with a uniform treatment distribution, which is independent of the covariate, provides randomized treatment samples for training a CEVAE. A uniform treatment selection process decouples $Z$ and $A$, thereby making $A$ independent of the covariate $X$, i.e. $p(A|X) = p(A)$. This is similar to a randomized clinical trial over treatment $A$ in Section 3.1. For this reason, it may be beneficial to train a CEVAE using a uniform treatment distribution. Here, the observational data-based distribution is $p(X, Y, A) = p(A|X)p(X)p(Y|X, A)$ and the corresponding uniform treatment distribution is $r(X, Y, A) = r(A)p(X)p(Y|X, A)$.

---

[9] A spillover happens when something in one situation affects something else in a different situation, even though they may not seem related.

jiwoong Im et al. (2021) use importance weighting to write the variational lower bound objective under the uniform treatment distribution (jiwoong Im et al., 2021):

$$\mathcal{L}_{\text{UTVAE}} = \mathbb{E}_{p(X,Y,A)}\left[w(X,A)\mathbb{E}_{q_\phi(Z|X,Y,A)}\left[\log\frac{p_\theta(X,Y,A|Z)p(Z)}{q_\phi(Z|X,Y,A)}\right]\right],$$

where $w(X,A) = \frac{r(A|X)}{p(A|X)} = \frac{1}{2p(A|X)}$ is the importance weight. $\frac{r(A|X)}{p(A|X)} = \frac{r(X,Y,A)}{p(X,Y,A)}$ and $r(A|X) = r(A) = \frac{1}{2}$, because of the independence between $X$ and $A$ in the causal graph and the uniformly distributed treatment selection procedure. UTVAE generalizes better than CEVAE especially when there is a distribution shift between training and inference. See the details of the training procedure of CEVAE and UTVAE for generative and inference networks respectively at (jiwoong Im et al., 2021).

### 4.5.3 Combining active and passive methods

So far, we have discussed active and passive CI learning separately. It is however often necessary to combine active and passive approaches in order to mitigate the bias arising from non-randomized data. Here, we review some of the methods that combine the two.

Sawant et al. (2018) use a bandit method to collect data online and estimate the ATE offline using the potential outcome model (Sawant et al., 2018). At each time step, they sample data using a bandit algorithm like Thompson sampling and aggregate a dataset (see Algorithm 5). Using this dataset, they update the model and re-estimate the potential outcome (see Algorithm 6). This approach combines active and passive learning since the model is updated in a batch training setting while the dataset can be collected asynchronously. Similarly, Ye et al. (2020) proposes a framework that combines the two methods. Here they use inverse probability weighting and matching algorithms for passive learning and UCB and LinUCB (Slivkins, 2019)) for active CI algorithms (Ye et al., 2020). In the inference time, Ye et al. (2020) estimate the potential outcomes using a passive method given a new unseen example $X = x_t$ for every action. If the new data point $(X = x_t, A = a_t)$ with an action $a_t$ is not in a non-randomized dataset, then the algorithm resorts to exploring the action for $X = x_t$ and adds the new data point $(X = x_t, A = a_t, Y = y_t)$ to the non-randomized dataset.

---

**Algorithm 5** Online Scoring and Batch training

Iteration $t = 0$; Log $L = \{\}$; contextual distribution parameter $\theta_0$ and $\theta_1$.
**for** $i = 1, 2, \cdots, T$ **do**
    **for** $t = 1, 2, \cdots, T$ **do**
        Sample data $x_t$
        Predict $\hat{Y}(1) = f(x_t, \theta_1)$
        Predict $\hat{Y}(0) = f(x_t, \theta_0)$
        Choose action $a_t = \text{argmax}_{a\in\{0,1\}}\hat{Y}(a)$
        Compute $p_t = p(a_t|x_t)$
        $L = L \cup (x_t, a_t, p_t)$
    **end for**
    Update $\theta_0$ and $\theta_1$ using Algo 6
**end for**

---

**Algorithm 6** Offline Batch Training

Dataset $D = \{\}$.
**for** $i = 1, 2, \cdots, T$ **do**
    Sample $(x_i, a_i, p_i, y_i(a_i))$ from $L$
    $\hat{y}_i(a_i) = y_i(a_i)/p_i$ - bias correction
    $\hat{y}_i(\neg a_i) = 0$
    **for** $m = 1, 2, \cdots, M$ **do**
        Sample $(x_m, \neg a_i, p_m, y_m(\neg a_m))$ from $L$
        $\hat{y}_i(\neg a_i) \leftarrow \hat{y}_i(\neg a_i) + \frac{1}{M}y_m(\neg a_i)/p_m$
    **end for**
    $\text{CATE}_i = \hat{y}_i(a_i) - \hat{y}_i(\neg a_i)$
    $D = D \cup (x_i, a_i, \text{CATE}_i)$
**end for**
update $\theta_0$ and $\theta_1$ by maximizing the likelihood on $D$.

---

## 5 Conclusion

The objective of this paper is to introduce various algorithms and frameworks in causal inference by categorizing them into passive and active algorithms. We have particularly focused on estimating average treatment effect (ATE), after outlining the standard assumptions necessary for the identification of causal effects: positivity, ignoreability, conditional exchangeability, consistency and Markov assumptions.

We first present the randomized controlled trial (RCT) as a representative example of an active causal inference algorithm. We then delve into bandit approaches that aim to balance the outcome itself and the quality of estimating the treatment effect. We explore different contextual bandit algorithms by considering various causal graph scenarios such as taking account of non-confounding variables or dealing with unknown confounding variables.

We then move on to discussing passive CI methods, including matching, inverse probability weighting and doubly robust methods. We touched upon deep learning-based approaches as well. A majority of these studies focus on converting a causal estimand into a statistical estimand, and subsequently, estimating the statistical estimand in order to obtain the causal estimate. These classical methods unfortunately do not perform well when they are put to work with high dimensional data. In order to overcome this challenge, deep learning has been proposed as a way to learn a compact representation suitable for estimating ATE. We thus discussed several deep learning-based approaches that learn to infer causal effects by automatically extracting hidden or unknown confounders' representations in problems with high dimensional data.

After reading the main part of this paper, readers should notice a clear resemblance between offline policy evaluation and passive causal inference methods. This resemblance is due to the similarity in estimating the reward in policy evaluation and estimating the potential outcomes in causal inference (Swaminathan & Joachims, 2015; Li, 2015). For example, some of the methods discussed in Section 4.5.3 have been used for offline policy evaluation in contextual bandit algorithms (Li et al., 2012; Sawant et al., 2018). Although we do not explore this connection further here, this is an important avenue to pursue both causal inference and reinforcement learning.

This review of causal inference is limited in two ways. First, we do not consider collider bias, and second, we assume a stationary conditional probability distribution. We discuss these two limitations briefly before ending the paper.

Collider bias happens when there is an extra variable, caused by both action and outcome variables, that is observed to be (conditioned on) a particular value. For example, suppose we investigate the effect of a new versus old medication on the patient outcome, and we gather data from a hospital. We divide the patients into two groups: those who received both the new and old medications. The study finds that the patients who received the old medication had better outcomes than the new one and the hospital conclude that the old medication is more effective on patient outcomes. It turns out this conclusion has a collider bias. This is because the decision to give the medication is based on the patient's recovery rate, which encourages doctors to prescribe a more potent drug. Thus, patients are more likely to receive the old medication which is stronger. In this case, the patient's recovery rate becomes a collider variable because it is caused by both the decision to give which medication and the patient's outcome. this is very separate from confounder bias and it doesn't address the collider bias issue

Besides collider bias, we have assumed that all conditional distributions are stationary (i.e. do not change over the course of causal inference and data collection). The question is what happens if these conditional distributions shift over time? One can introduce temporal dependencies to a causal graph to describe such shifts over time. Still, it is challenging to work with such a graph because collected data points over time are correlated with each other. This violates the no interference/Markov assumption we discussed earlier in Section 2.3. To address this problem, various time series methods have been developed that take into account the temporal dependence of the data. For instance, deep sequential weighting (DSW) and sequential causal effect variational autoencoder (SEVAE) estimate ITE with time-varying confounders (Trifunov et al., 2022; Liu et al., 2020; Kumor et al., 2021).

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

# A   Appendix

## A.1   Naive ATE estimation

Having an RCT dataset, by default gives us

$$\{Y(0), Y(1)\} \perp\!\!\!\perp A | X = x, \text{ for all } x \in \mathcal{X}.$$

We can compute ATE by aggregating differences in mean estimators of treatment and control group

$$ATE = \mathbb{E}_{p(x)}[\mathbb{E}[Y(1) - Y(0)|X = x]]$$
$$= \mathbb{E}_{p(x)} [\mathbb{E}[Y(1)|X = x] - \mathbb{E}[Y(0)|X = x]]$$

Taking the Monte Carlos approximation, we have

$$\widehat{ATE} = \mathbb{E}_p(x) \left[ \frac{1}{N_{x1}} \sum_{i:t_i=1} y_i - \frac{1}{N_{x0}} \sum_{j:t_j=0} y_j \right]$$
$$= \frac{N_x}{N} \sum_x \left[ \frac{1}{N_{x1}} \sum_{i:t_i=1} y_i - \frac{1}{N_{x0}} \sum_{j:t_j=0} y_j \right]$$

where $N_{xt}$ is the number of $x$ data points with treatment assignment $t$, $N_x$ is number of $x$ data points, $N$ is the total data points. For simplicity, let $\bar{e}(x) = N_{x1}$ and $1 - \bar{e}(x) = N_{x0}$. We can re-express the $\widehat{ATE}(x)$ as

$$\widehat{ATE}_{\text{AGG}}(x) = \frac{1}{\bar{e}(x)} \sum_{i:t_i=1} y_i - \frac{1}{1 - \bar{(e)}(x)} \sum_{j:t_j=0} y_j.$$

Our $\widehat{ATE}(x)$ is unbiased estimator and the estimation error is

$$\sqrt{N_x}(\widehat{ATE}_{\text{AGG}}(x) - ATE(x)) \rightarrow \mathcal{N} \left( 0, \frac{\text{Var}[Y(1)|X = x]}{\bar{e}(x)} + \frac{\text{Var}[Y(0)|X = x]}{1 - \bar{e}(x)} \right).$$

Under the assumption that $\mathrm{Var}[Y(A)|X = x] = \sigma^2(x)$ does not depend on $A$, then we

$$\sqrt{N_x}(\widehat{ATE}_{\mathrm{AGG}}(x) - ATE(x)) \to \mathcal{N}\left(0, \frac{\sigma^2(x)}{\bar{e}(x)(1 - \bar{e}(x))}\right).$$

We can re-write our estimator by decomposing in terms of true ATE and the approximation error:

$$\widehat{ATE}_{\mathrm{AGG}} = \sum_x \frac{N_x}{N} \widehat{ATE}_{\mathrm{AGG}}(x)$$

$$= \sum_x p(x) ATE(x) + \underbrace{\sum_x p(x)(ATE(x) - \widehat{ATE}(x))}_{\approx \mathcal{N}(0, \sum_x p(x)^2 \mathrm{Var}[\widehat{ATE}(x)]}$$

$$+ \underbrace{\sum_x \left(p(x) - \frac{N_x}{N}\right) ATE(x)}_{\approx \mathcal{N}(0, N^{-1}\mathrm{Var}[ATE(x)]} + \underbrace{\sum_x \left(p(x) - \frac{N_x}{N}\right)(ATE(x) - \widehat{ATE}(x))}_{= \mathcal{O}(N^{-1})}.$$

This makes $\widehat{ATE}_{\mathrm{AGG}}$ error to distribute $\mathcal{N}(0, \mathrm{Var}_{AGG})$, where

$$N\mathrm{Var}_{AGG} = \mathrm{Var}[ATE_{\mathrm{AGG}}(x)] + \mathbb{E}_p(x)\left[\frac{\sigma^2(x)}{\bar{e}(x)(1 - \bar{e}(x))}\right]. \tag{16}$$

## A.2 ATE ordinary least-squares estimator

Rather than taking the difference in direct estimations of expected potential outcomes, we will model the potential outcome using linear regression,

$$Y_i(t) = c_t + X_i \beta_t + \epsilon_i(t)$$

where $\mathbb{E}[\epsilon_i(t)|X_i] = 0$ and $\mathrm{Var}[\epsilon_i(t)|X_i] = \sigma^2$. The mean of data to be zero $\mathbb{E}[X] = 0$ by normalizing the dataset and the variance of data is $\mathrm{Var}[X] = \sigma_X$. Because we are in RCT setting, $p(T = 1) = p(T = 0) = \frac{1}{2}$.

In this setup, we can write the ATE as

$$ATE = \mathbb{E}[Y(1) - Y(0)] = c_1 - c_0 + \mathbb{E}[X](\beta_1 - \beta_0).$$

Now we can run ordinary least-square (OLS) estimator to estimate $c_t$ and $\beta_t$,

$$\hat{\tau}_{OLS} = \hat{c}_1 - \hat{c}_0 + \bar{X}(\hat{\beta}_1 - \hat{\beta}_0),$$

where $\bar{X} = \frac{1}{n} \sum_{i=1}^n x_i$. The standard error for $c_t$ and $\beta_t$ estimation are

$$\hat{c}_1 - c_1 = \mathcal{N}\left(0, \frac{\sigma^2}{n_1}\right) \text{ and } \hat{c}_0 - c_0 = \mathcal{N}\left(0, \frac{\sigma^2}{n_0}\right)$$

respectively. Since $\mathbb{E}[X] = 0$ and $\bar{X}$ asymptotically approaches to 0 with the standard error of $\|\beta_1 \beta_0\|_{\sigma_X}^2 / N$, we have

$$\hat{\tau}_{OLS} - \tau = \hat{c}_1 - c_1 - \hat{c}_0 - c_0 + \bar{X}(\beta_1 - \beta_0) + \bar{X}(\hat{\beta}_1 - \beta_1 - \hat{\beta}_0 + \beta_0)$$

where the last term has $O(1/n)$ error rate. This makes $\widehat{ATE}$ error to distribute $\mathcal{N}(0, \mathrm{Var}_{OLS})$, where $\mathrm{Var}_{OLS}) = 4\sigma^2 + \|\beta_0 - \beta_1\|_{\sigma_X}^2$.

### A.3 Active CI Learning

### A.3.1 Contextual Bandit Algorithm

Contextual bandit is known as an online method that helps you make decisions in given contexts. It finds optimal decisions by maximizing their total rewards over a period of time. In contextual bandit setup, we get an observation $x_t$ at time $t$, the algorithm picks an action from a finite set $a_t \in \mathcal{A}$, and then executes action $a_t$ on observation $x_t$. The reward $y_t \in [0, 1]$ is given by the world which is some distribution parameterized by $(X, A)$ variables and the samples are drawn independently and identically. The reward depends on both the context $x_t$ and the chosen action $a_t$ at each round. The reward distribution can change over time but this change is explained by the stream of context data[10]. The action is chosen based on the choice of algorithm, such as UBC1 and Thomson sampling methods Agrawal & Goyal (2013)[11] (see Appendix A.3.1).

**Examples of contextual bandits' policy algorithms**

---
**Algorithm 7** Epsilon-Greedy

---
Toss a coin with success probability $\epsilon_t$.
**if** success **then**
    explore: choose an arm uniformly at random.
**else**
    exploit: choose the arm with the highest average reward so far.
**end if**

---

---
**Algorithm 8** "High-confidence elimination"

---
Alternate two arms until $UCB(a_t) < LCB(a'_t)$ after some even round t.
Abandon arm $a$, and use arm $a'$ forever since.

---

---
**Algorithm 9** UBC1

---
pick arm some a which maximizes $UCB(a_t)$

---

---
**Algorithm 10** Thompson Sampling

---
Sample mean reward vector $\mathbb{E}[Y(a)]$ for each action $a$ from the posterior distribution $p(a|X)$.
Choose the best arm $a_t$ according to $\mathbb{E}[Y(a_t)]$.

---

**Upper bound on ATE**

Both *high-confidence elimination method* and *UCB* algorithms have assured to be upper-bounded for ATE estimations since the potential outcomes are distanced enough that the two confidence intervals do not overlap,

$$ATE = |\mathbb{E}[Y(A = a)] - \mathbb{E}[Y(A = a')]| \leq 2(r_t(a) + r_t(a')) \leq 4(\sqrt{2\log(T)/\lfloor t/2 \rfloor}) \leq O(\sqrt{\log(T)/t}).$$

where $T$ is the total iterations and $n_t(a) = \lfloor \frac{t}{2} \rfloor$ since $a$ and $a'$ has been altered. In this case, we can aggregate data into an intervention dataset until the confidence interval does not overlap and use them to estimate ATE.

---
[10]The reward and outcome can be viewed the same and used interchangeably.
[11]We will not review the details of each algorithm in this paper but refer the reader to Bouneffouf et al. (2020).

### A.4 Passive CI Learning

#### A.4.1 Inverse Probability Weighting

Since propensity score $e(X)$ is unknown in practice, we have to estimate $\hat{e}(X)$ via parametric and non-parametric regression. We use non-parametric regression to estimate $\hat{e}(X) = \frac{N_1}{N}$ where $N_1 = \sum^{i=1} \mathbb{I}[Y(X) = 1]$. ATE estimation is

$$\widehat{ATE}_{\text{IPW}} = \frac{1}{N} \sum_i^n \left[ \frac{\mathbb{I}[A_i = 1] y_i(1)}{\hat{e}(x_i)} \right] - \frac{1}{N} \sum_i^n \left[ \frac{\mathbb{I}[A_i = 0] y_i(0)}{1 - \hat{e}(x_i)} \right].$$

Suppose that $\hat{e}(x) \to e(x)$ as $N \to \infty$ (i.e., $\sup_{x \in \mathcal{X}} |e(x) - \hat{e(x)}| \to \mathcal{O}(a_n)$). Then, the ATE error becomes

$$|ATE - \widehat{ATE}_{\text{IPW}}| = \mathcal{O}\left( \frac{a_n}{\eta} \right)$$

where $\eta \leq e(x) \leq 1 - \eta$ for all $x \in \mathcal{X}$ and $|Y_i| \leq 1$. Therefore, $\widehat{ATE}$ is concentrated at $\frac{1}{\sqrt{n}}$.

The weighted population outcomes of two actions, $A = 0$ and $A = 1$, is an unbiased estimate of the average treatment effect

$$ATE = \mathbb{E}\left[ \frac{\mathbb{I}[A = 1] Y(1)}{e(X)} \right] - \mathbb{E}\left[ \frac{\mathbb{I}[A = 0] Y(0)}{1 - e(X)} \right]. \tag{17}$$

In order to express the variance of $ATE$ in Equation 17, let us re-express $Y(0)$ and $Y(1)$,

$$Y(0) = c(X) - (1 - e(X)) ATE(X) + \epsilon(0)$$
$$Y(1) = c(X) + e(X) ATE(X) + \epsilon(1)$$

where $c(X)$ is a function that makes the expression above work, and $\mathbb{E}[\epsilon(0)|X] = 0$ and $\mathbb{E}[\epsilon(1)|X] = 0$. assume that $\text{Var}(\epsilon(A)|X) = \sigma^2(X)$ does not depend on $A$. Then

$$\begin{aligned}
N\text{Var}_{IPW}[ATE(X)] =& \text{Var}\left[ \frac{AX}{e(X)} - \frac{(1-A)Y}{1-e(X)} \right] \\
=& \text{Var}\left[ \frac{Ac(X)}{e(X)} - \frac{(1-A)c(X)}{1-e(X)} \right] + \text{Var}[ATE(X)] + \text{Var}\left[ \frac{A\epsilon(X)}{e(X)} - \frac{(1-A)\epsilon(X)}{1-e(X)} \right] \\
=& \mathbb{E}\left[ \frac{c(X)^2}{e(X)(1-e(X))} \right] + \text{Var}[ATE(X)] + \mathbb{E}\left[ \frac{\sigma^2(X)}{e(X)(1-e(X))} \right]
\end{aligned}$$

Note that we can express the variance of IPW estimator in terms of the variance of our estimator is worse than the variance of aggregating difference in mean estimator $\text{Var}_{AGG}$ in Equation 16, which is the naive ATE estimator for RCT dataset,

$$N\text{Var}_{IPW}[ATE(X)] = N\text{Var}_{AGG}[ATE(X)] + \mathbb{E}\left[ \frac{c(X)^2}{e(X)(1-e(X))} \right].$$

This means that IPW has higher variance than AGG estimator. Surprisingly, we can conclude that the true propensity score performs worse than empirical propensity score, since AGG estimator used $\bar{e}(x) = \frac{N_x}{N}$ (see Section A.1).

#### A.4.2 Domain Invariance Regularization

Intuitively, inducing the treated and control representational distribution to be the same is that it induces the two learned prediction function $p_\theta(y|t = 0, x)$ and $p_\theta(y|t = 1, x)$ to have better generalization across the

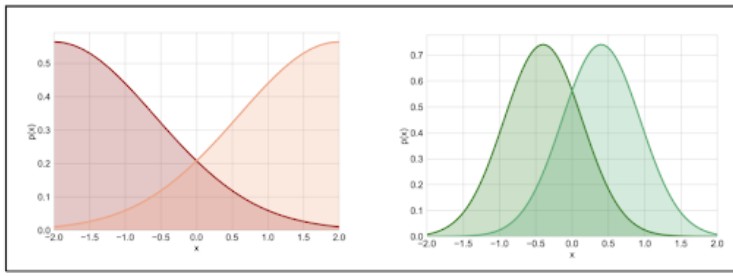

Figure 6: Red distribution (left) has large overlap on the tails and the green distribution has small overlap on the tails

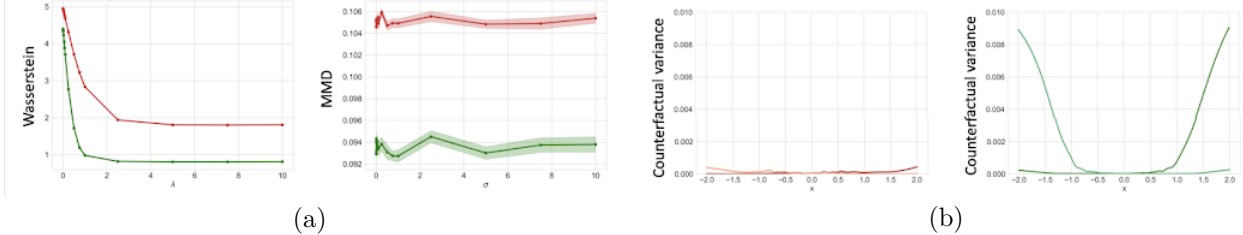

(a)                                                                            (b)

Figure 7: (a) Results using IPMs - Wasserstein distance and MMD, (b) Results using counterfactual variance

treated and control populations. Indeed, Shalit et al. (2017) show that CFR objective function is the upper bounds of the PEHE generalization error Bareinboim & Pearl (2016),

$$PEHE \leq 2(\mathcal{L}_\mathrm{F} + \mathcal{L}_\mathrm{CF} - 2\sigma_Y^2) \leq 2(\mathcal{L}_\mathrm{F}^{T=0} + \mathcal{L}_\mathrm{F}^{T=1} + B_h \mathrm{IPM}_\mathcal{F}(\hat{p}_{T=1}(h(X)), \hat{p}_{T=0}(h(X)))$$

where the expected factual and counterfactual losses are defined as

$$\mathcal{L}_\mathrm{F} = \mathbb{E}_{p(X,T)}[l(X,T)] = u\mathcal{L}_\mathrm{F}^{T=1} + (1-u)\mathcal{L}_\mathrm{F}^{T=0}$$
$$\mathcal{L}_\mathrm{CF} = \mathbb{E}_{p(X,1-T)}[l(X,T)] = (1-u)\mathcal{L}_\mathrm{CF}^{T=0} + u\mathcal{L}_\mathrm{CF}^{T=1}$$

respectively. The expected loss function for individual data point over $p(Y_T|X=x)$ is $l(X=x, T=t)$ and $u = p(T=1)$ is the proportions of treated in the population. The expected factual/counterfactual treated and control losses becomes

$$\mathcal{L}_\mathrm{F}^{T=1} = \mathbb{E}_{p(X,1)}[l(X,1)], \qquad \mathcal{L}_\mathrm{F}^{T=0} = \mathbb{E}_{p(X,0)}[l(X,0)]$$
$$\mathcal{L}_\mathrm{CF}^{T=1} = \mathbb{E}_{p(X,0)}[l(X,1)], \qquad \mathcal{L}_\mathrm{CF}^{T=0} = \mathbb{E}_{p(X,1)}[l(X,0)]$$

respectively. $\sigma_Y^2 := \min\{\sigma_0^2, \sigma_1^2\}$ and $\sigma_t^2 = \mathbb{E}_{p(X,T)}[(Y - f_\phi(X))^2]$ is the expected variance of $Y_T$. The full proof can be found in the original paper Shalit et al. (2017) but the key idea is that $\mathcal{L}_\mathrm{CF} \leq u\mathcal{L}_\mathrm{F}^{T=0} + (1-u)\mathcal{L}_\mathrm{F}^{T=1} + B_h \mathrm{IPM}_\mathcal{F}$.

### A.5 Posterior Variance Reduction Regularization

Yao et al. (2020) demonstrate why distributional distance to balance the treated and controlled representations is not ideal using an toy example in Figure 6. The red population comes from two truncated normal distributions having large overlap in tails and the green population comes from two normal distributions having small overlap in the tails. Figure 7(a) illustrates that both the MMD Gretton et al. (2012) and Wasserstein distances Villani (2009) are smaller in the green population compared to the red population, even though sufficient support is satisfied in the red population and not the green population. In contrast, counterfactual variance perfectly describes the lack of support in the red population as shown in Figure 7(b).

Minimizing the counterfactual variance can lead to better generalization error Yao et al. (2020). The follow Theorem shows that the counterfactual Gibbs risk $R_{p(1-T)}$ is upper bounded by two terms that corresponds

to domain invariance and counterfactual variance,

$$R_{p(1-t)} \leq \sup_x \frac{p(x, 1-t)}{p(x, t)} \mathcal{L}_{\text{Factual}} + \frac{1}{2} \mathbb{E}_{p(x)}[\sigma^2(x|\mathcal{X}, \theta)].$$

We observe that the first term consists of the factual loss and distribution mismatch. Minimizing both factual loss and the making posterior distribution to be invariant will lead to lower counterfactual Gibbs risk. The second term corresponds to counterfactual variance. This illustrates that minimizing the counterfactual variance is indispensable regularization term as well.

## A.6 Deep Latent-Variable Model: UTVAE

In the CEVAE, there are two conditional distributions that depend on treatment $T$, $p_\theta(Y|T, Z)$ and $q_\phi(Z|T, X, Y)$. Both of these distributions can be estimated using samples drawn from a treatment distribution that is either dependent on or independent of the confounding factor. In doing so, we have the option to use observational data based, or uniform treatment distributions, for estimating generative and inference distributions respectively

$$\mathcal{L}(\theta; \phi) = \mathcal{L}_{\text{CEVAE}}(\theta; \bar{\phi}) + \mathcal{L}_{\text{UTVAE}}(\phi; \bar{\theta})$$

where $\bar{\theta}$ and $\bar{\phi}$ are fixed parameters - the gradients with respect to these variables are blocked in the computational graph. We do so in order to isolate the impact of the choice of treatment distribution on the associated conditional distributions.

