# OpenReview forum: "Active & Passive Causal Inference: Introduction"
_TMLR — Rejected by TMLR_

### Review · Reviewer_56Z6 · 2024-01-22

**Summary Of Contributions:**

The paper titled "Active & Passive Causal Inference" provides a comprehensive introduction to causal inference, particularly aimed at machine learning researchers, engineers, and students who are new to this field. It categorizes causal inference methods into two main groups: active and passive. The paper discusses several crucial assumptions necessary for causal identification, such as exchangeability, positivity, consistency, and the absence of interference, and explains how these assumptions underpin different causal inference techniques.

In active causal inference, the paper covers methods like randomized controlled trials (RCTs) and bandit-based approaches. These methods actively intervene in the data collection process, with RCT being a standard approach in fields like clinical trials. The paper also reviews bandit methods for causal inference, which offer a more sophisticated way to estimate average treatment effects (ATE) by occasionally randomizing actions based on the chosen bandit algorithm.

In contrast, passive causal inference methods work with datasets that have been collected without active intervention. The paper discusses classical approaches like matching, inverse probability weighting, and doubly robust methods, as well as more recent developments involving deep learning.

**Audience:**

Yes

**Broader Impact Concerns:**

There is no ethical concern as far as I know.

**Claims And Evidence:**

Yes

**Requested Changes:**

1. Clarification and simplification for beginners: To better cater to the target readers of beginners, the paper should include an explanation of new concepts and jumping steps about causality, such as the calculation of $p(Y = y| do(A=a))$ and the definition of causal graph. The paper seems to use causal graph without introducing the definition of  causal graph.
2. Expansion on Advanced Topics: While the paper does an excellent job of covering traditional methods in causal inference, it falls short in exploring more advanced techniques, particularly those beyond basic deep learning methods. To provide a more rounded perspective, it would be beneficial to include a section or a discussion on recent advancements and trends in causal inference, highlighting how these new methods compare or integrate with the traditional approaches.
3. Correction of Linguistic and Typographical Errors: The paper currently contains several linguistic and typographical errors that need to be addressed to improve its readability and professionalism.

**Strengths And Weaknesses:**

Strengths:

1. **Comprehensiveness:** The paper provides a thorough overview of both active and passive approaches to causal inference, making it a valuable resource for beginners in this field.
2. **Clarity in Assumptions:** It clearly outlines the fundamental assumptions required for causal inference, offering a solid foundation for understanding the subject.

Weaknesses:

1. Although it is targeted at beginners, some parts may be too specialized or assume prior knowledge, which can be challenging for the intended audience. For example, the article directly gives the calculation formula $p(Y = y| do(A=a)) = \sum_x p(X = x)p(Y = y | A = a, X = x)$, but does not explain why this formula calculates the probability of $Y = y$ when $A = a$ is enforced;
2. Limited exploration of advanced topics: The paper seems to focus more on traditional methods, rather than on newer or more advanced causal inference techniques beyond basic deep learning methods.
3. Linguistic and Typographical Errors: The paper contains several errors that detract from its overall quality, such as:

   a. "There are however other methods in the same category that **aims** ..."
   b. "The first such assumption is **exchangeabilty** which ..."
   c. "... the expected potential outcome of counteraction being higher **than of** action ..."
   d. "Such an algorithm generates a series of matched sets that **consists** of ..."
   e. "The propensity score theorem tells us that if **ignoreability** is satisfied ..."
   f. "... either the propensity score or the estimator µA(X) **are** correct ..."

---

> ### Author Response · Authors · 2024-02-25
> **Thank you for noticing that this paper is valuable for the beginners**
>
> We appreciate your diligent efforts in reviewing our paper. We are happy to hear that the paper is comprehensive and outlines and fundamental assumptions are clear. Also, thank you for noticing that this paper is valuable for the beginner in this field!
>
> **Comment1**: *"The article directly gives the calculation formula  $p(y=do(A=a))= \sum_x p(Y|A=a,x)p(x)$, but does not explain why this formula calculates the probability of Y=y when A=a is enforced"*
>
> RE: $P(Y|A=a,x)$ tells us the effect of A on Y given X. $P(Y|do(A=a))$ tells us the effect of A on Y over the entire population where it measures Y when A=a was forcefully assigned after marginalizing out X). We added an additional description.
>
>
> **Comment 2**: *The paper seems to use a causal graph without introducing the definition of a causal graph.*
>
> RE:  A causal graph is a graphical representation of causal relationships among variables in a system. It visually depicts how variables influence each other, helping us to understand and analyze the causal structure of a phenomenon.
> Figure 1 shows the graph representation of causal relationships for a variety of CI methods. Depending on the data collection process and experimental setup, certain methods do not require knowing the causal graph in priority, e.g., RCT, while other methods require knowing the structure of a causal graph. We added the following casual graph description on page 4.
>
>
> **Comment 3**: *It falls short in exploring more advanced techniques, particularly those beyond basic deep learning methods. To provide a more rounded perspective, it would be beneficial to include a section or a discussion on recent advancements and trends in causal inference, highlighting how these new methods compare or integrate with the traditional approaches.*
>
> RE: While there are indeed numerous advanced papers in the field of causal inference, covering all advances comprehensively may exceed the scope of this literature review, particularly given its emphasis on providing a beginner-friendly introduction. However, we acknowledge the importance of pointing readers towards the future directions of causal inference.
>
> Section 4.5 consists of recent trends in CI research. To address your comment, we will integrate additional recent advancements and the evolving landscape of causal inference in Section 4.5.
>
>
>
> **Comment 4**: *Linguistic and Typographical Errors: The paper contains several errors that detract from its overall quality, such as:
> a. "There are however other methods in the same category that aims ..."
> b. "The first such assumption is exchangeabilty which ..."
> c. "... the expected potential outcome of counteraction being higher than of action ..."
> d. "Such an algorithm generates a series of matched sets that consists of ..."
> e. "The propensity score theorem tells us that if ignorability is satisfied ..."
> f. "... either the propensity score or the estimator µA(X) are correct ..."*
>
> RE: thank the reviewer for noticing these typos and grammatical errors. We fixed them accordingly:
> - Aims -> Aim
> - exchangeabilty ->exchangeability
> - than of -> than
> - consists -> consist
> - ignoreability -> ignorability
> - are -> is

---

### Review · Reviewer_RLMu · 2024-01-29

**Summary Of Contributions:**

This paper aims at being an introduction to algorithms that solves active and passive causal inference. In both cases, the algorithm tries to estimate the effect of a given action $A\in\{0,1\}$.  The passive case corresponds to an on offline data where the actions where not chosen by the algorithm. The active case corresponds to a situation where the algorithm *actively* choose the actions as a function of what it wants to estimate. After recalling major definitions, the authors first cover the active case, which seems a relatively mature field, and expose the  link with contextual bandits problems. The second topic then cover the topic of passive causal interference, that seems highly dependent on the assumptions that are made on data. This part is more heuristic.

**Audience:**

Yes

**Broader Impact Concerns:**

I do not foresee direct impact of the work.

**Claims And Evidence:**

No

**Requested Changes:**

The "weakness" part above indicates a lot of clarifications that could be made.

**Strengths And Weaknesses:**

(I must admit that I am not an expert of the field to this review should be taken as the view of an outsider. Since this paper is supposed to be an 'introduction', this might reflect the feeling of some of the readers of the paper).

The paper is relativey easy to read but below are some comments (in order of appearance):

Section 2 (background): this part sets a few definitions that are going to be used. I think that it is relatively clear but I think that the authors should be more precise and consistent concerning the vocabulary that is used. In particular, there seems to be a confusing in the paper between "confoundedness, confoundness and ignorability/exchangeability. To me, this seems to be a central notion in causal inference and I think that it should be explained more clearly. Also, in general the paper should be more precise about the definition and assumptions that are used. For instance: it is not clear what the "big formula" in page 7 is: ATE is "computed as follows" if we assume the above assumptions but why and when should we assume them?

Section 3 (active inference): I quite liked this part and I found it clear and relatively well documented. Most of the presented material seems classical and not very recent but the link with contextual bandit is interesting.

Section 4 (passive inference): This seems to be the major part of the paper but I must say that I found this part very confusing. For instance:
- in Section 4.2 and 4.3, it is very unclear to me how the effect of confunder can be ignored (or not) and under which assumptions do the methods presented here work. For instance: is the matching robust to the existence of confunder variables? (I would say not but the writing seems to indicate the opposite).
- in Section 4.3, I do not get why one estimator is biased and the next is not. Also, at some point a variable "follows a Normal distribution" (page 16). Why? Is it an assumption or a consequence?
- The "modern" part of the paper seems to be in section 4.5. For an outsider, I found this part very hard to read and I am not sure what information I should gain from this.

Minor remarks:
- there are a few references to an Appendix that are "??"
- Figure 1 and 4 are (almost) identical. Could this be better used?

---

> ### Author Response · Authors · 2024-02-25
> **Happy to hear that you quite like the Section 3: Active CI learning!**
>
> We sincerely thank you for your great efforts in reviewing the paper. We are glad that section 3: active causal inference is clear and well-documented! We addressed your concerns about section 4 and added additional explanations. Please see the detailed response below:
>
> **Comment 1**: *"Section 2 (background): … In particular, there seems to be a confusing in the paper between "confoundedness, confoundness and ignorability/exchangeability. To me, this seems to be a central notion in causal inference and I think that it should be explained more clearly. "*
>
> RE: Ignorability/exchangeability is an assumption while unconfoundedness is a goal to be obtained by the experimentalist. To meet the ignorability assumption, we need to achieve unconfoundedness, which refers to having an absence of confounding variables in a causal relationship. This involves careful design, data collection, and measurements to control potential confounders and isolate the impact. In Section 2.2, we distinguish these two concepts.
>
>
> **Comment 2**: *it is not clear what the "big formula" in page 7 is: ATE is "computed as follows" if we assume the above assumptions but why and when should we assume them?*
>
> RE: The ATE is a statistical quantity and is only a causal quantity under the assumptions above. The derivation in page 7 shows how each assumption is applied to go from causal estimand to statistical estimand. We made an edit to the text and described why assumptions are needed. We will be more clear and precise about the assumptions.
>
> **Comment 3**: *In Section 4.2 and 4.3, it is very unclear to me how the effect of confounder can be ignored (or not) and under which assumptions do the methods presented here work.*
>
> RE: When we say the effect of a confounder is "ignored," it means that these methods aim to eliminate or reduce the influence of confounding variables on the estimation of causal effects. This is achieved by either matching treated and control units based on their covariates (matching) or weighting observations to balance the distribution of covariates between treatment groups (inverse probability weighting). Under positivity and ignorability assumptions, the methods presented in 4.2 and 4.3 can effectively address confounding bias.
>
>
> **Comment 4**: *Is the matching robust to the existence of confunder variables? (I would say not but the writing seems to indicate the opposite).*
>
> RE: Matching method removes confoundedness by creating comparable groups based on observed covariates.  It selects individuals from different treatment groups but has similar characteristics or covariate distributions. This ensures that the treatment and control groups are balanced concerning the potential confounders. We made edits accordingly in the paper.
>
> **Comment 5**: *"In Section 4.3, I do not get why one estimator is biased and the next is not. Also, at some point a variable "follows a Normal distribution" (page 16). Why? Is it an assumption or a consequence?"*
>
> RE: ATE_AGG is a naive estimator from the first equation in section 4.1. The naive ATE_AGG estimates the conditional probability of P(Y|A) instead of P(Y|do(A)). ATE_AGG is biased since P(Y|A) is seen as the biased version of P(Y|do(A)).  We believe the confusion may have arisen from our use of \hat{e} in \hat{ATE}_{AGG}. We will replace \hat{e} with \eta.
>
> The estimation error follows a Normal distribution with zero mean and some variance because it is estimated using Monte Carlo sampling and the sum of i.i.d random variables approaches Normal distribution, and it has zero mean.
>
> **Comment 6**: *The "modern" part of the paper seems to be in section 4.5. For an outsider, I found this part very hard to read and I am not sure what information I should gain from this.*
>
> We appreciate the reviewer’s feedback regarding Section 4.5 and acknowledge your concerns about its clarity. This section intends to introduce modern advancements in CI, specifically focusing on the application of deep learning in CI. We understand that this content may be challenging for readers unfamiliar with deep learning. We will make edits to provide additional context and explanation for better comprehension.

---

### Review · Reviewer_th1G · 2024-02-12

**Summary Of Contributions:**

This paper is presented as an introduction to the field of causal inference for newcomers. It adopts the framework of potential outcomes from Donald Rubin, and presents a range of existing methods for estimating causal effects in contextual bandits, categorized into active and passive methods. There is no new knowledge presented in the paper, the contribution is mainly to present a collection of existing  works on causal inference in a unified framework, for a broad audience.

**Audience:**

No

**Broader Impact Concerns:**

No concern

**Claims And Evidence:**

No

**Requested Changes:**

Improve the overall consistency and mathematical soundness of the paper. It is possible to be clear, concise and appealing to newcomers without sacrificing correctness. Clarify and acknowledge other frameworks such as do-calculus, in particular if you use its notations like $do(A=a)$. Make it very clear, from the start, the context, the assumptions and the limitations of the paper, and of each of the presented methods method. There is no room for ambiguity in an introductory paper. The field of causal inference has ambiguity enough already, efforts are much needed in the other direction.

**Strengths And Weaknesses:**

While the intent of the paper is noble, I find it rather weak in achieving its purpose: introducing newcomers to the field of causal inference. First, it only considers a single framework, that of potential outcomes, and disregards other existing frameworks without even a mention. Then, I found several inconsistent statements and notations (see detailed comments), which made the presentation confusing even for me, someone with expertise in the field. It is also unclear when reading the paper what are contributions from the authors, and what are existing frameworks and algorithms (we introduce, we estimate etc. are used freely at several places), making the paper further ambiguous. The scope of the paper is also rather restricted (contextual bandits and potential outcomes), while the title is very general. Overall, I do not find this paper to bring anything of value to the field or the community, especially in light of the plethora of existing introductory works in the litterature [1-5].

[1] An Introduction to Causal Inference, Judea Pearl 2010

[2] Introduction to Causal Inference, Spirtes 2010

[3] Introduction to Causal Inference, Neal 2020

[4] Observation and experiment: An introduction to causal inference, Rosenbaum 2017

[5] Causal Inference, Ding & Li 2018

Detailed comments:

 - p1: "potential outcome" -> the concept of potential outcomes is well-established, here it seems the authors come up with it. A reference is missing.

 - p2: "In this paper [...] under such a causal graph." -> The introduction starts getting confusing here. Is it absolutely necessary to assume the availability of a causal graph to do CI? What does a causal graph allow for, which its absence does not allow? Until now it has not been said if the problem of interest was CI from interventional data, observational data, or a mix of both. The challenges of CI are not the same in each of these settings, and knowledge of the causal graph is not required in the purely interventional setting.

 - p2: "2b condition" -> typo?

 - p2: "The action A [...] outcome Y." -> I find the terms employed a bit confusing. Earlier we can read "what is the impact", now we have "causal effect", "outcome" and "affected" Do these terms refer to the same thing? I'd suggest sticking to a unique terminilogy in the paper to avoid confusion.

 - p2: "We measure [...] treatment effects." -> What are average and individual treatment effects? This paper claims to be a starting point for people interested in but not familiar with causal inference, yet here it assumes right-away that the reader is familiar with non-trivial concepts.

 - p2: There exist other frameworks for CI, not mentioned here, such as the do-calculus framework of Judea Pearl in which these assumptions all boil down to a simpler condition, identifiability. If this paper only covers the potential outcomes framework of Donald Rubin, then it should be clearly stated. Unfamiliar readers (the target audience of the paper) might be lured to believe potential outcomes is the one and only framework for CI.

 - p2: the classic causal graph in Figure 1 -> Figure 1 exposes several graphs, which one is referred to here? Also, is it meant here that bandit algorithms are learning a causal graph?

 - p2: RUBIN & THOMAS -> typesetting

 - p3: Figure 1 -> these are graphs, not CI methods. What is the meaning of squares vs circles, and dark grey vs light grey nodes? What is the difference between the "deep latent models" and "deep causal models" graphs ? Moreover, it is not clear to me what the methods "causal bandit" or "deep causal method" refer to here. References are missing.

 - p4: "We define the joint, conditional and intervention probabilities" -> Where does the intervention probability come from? This formula is valid only for specific graphs, which is not made clear from the text. Also, the do-calculus notation is used here, while only the potential outcome framework was referred to in the text earlier. This is inconsistent and confusing, especially for an introductory paper. Other information is missing, like p which is introduced without context or definition. Following what is said earlier in the introduction, which setting are we in here? Observational? Interventional? Passive? Active?

 - p5: where we marginalize out the covariate X -> X is also marginalized out in your expected potential outcome formula.

 - p5: individual as well as expected -> What is meant here by individual? A definition is missing.

 - p5: P -> is this capital P different from the lowercase p used earlier?

 - p5: This dataset does not, or perhaps more correctly cannot -> does not or can not? Your data distribution being iid, I do not see why counterfactual outcomes as defined here could not be obtained in a large enough dataset.

 - p8: $Y_{X}(A)$ -> This is inconsistent with Algorithm 1, where potential outcome is $Y(A)$. To this point, I am still unsure which setting is studied here. Are we in a bandit setting, where we are looking for an unconditional policy for $A$ ? Or are we in a contextual setting, where we want a policy for $A | X$ ?Why is Algorithm 1 sampling actions from a contextual policy, but then marginalizes out the covariate $X$ when computing the potential outcomes? I am very confused.

 - p8: there is a major difference -> I believe this is an overstatement. Most bandit algorithms keep track of the expected outcome of each action, in order to find the best policy. And, as stated in the introduction, CI models potential outcomes so that it can pick the action that maximizes the outcome. I hardly see any difference here.

 - p10: We then estimate the average treatment effect (ATE) -> I find it very confusing that the authors frequently say "We do X", "We propose X", "We introduce X", while they also claim this paper to be an introduction to existing methods. At times it feels like the presented methods and algorithms are the author's own contribution.

 - p10: appendix ?? -> broken ref

 - p23: We discuss these two limitations briefly before ending the paper -> These limitations, or rather assumptions, should be stated clearly at the beginning of the paper to give a proper context, not at the end.

---

> ### Author Response · Authors · 2024-02-25
> **[1/3] Thank you for noticing that the intent of the paper is noble!**
>
> Glad that the reviewer noticed our effort in the presentation and collections of existing works on causal inference in a unified framework, for a broad audience.
>
> Upon review of the reviewer's feedback:
>
> - We recognize the reviewer's concerns regarding the utilization of both potential outcomes and do-calculus frameworks in our paper. We clarify that we intend to provide a comprehensive and unified perspective on causal inference, aiming to bridge the intuitive conceptualization of potential outcomes with the formal treatment of intervention probabilities. While our initial focus on potential outcomes serves to illustrate treatment effects effectively, we acknowledge the necessity of incorporating intervention probabilities for a deeper understanding of causal models.
>
> - We acknowledge the reviewer's perspective on the perceived lack of novelty in our discussion of active causal inference and the application of bandit algorithms for data collection. However, we highlight the utilization of bandit methods for estimating causal effects in dynamic environments, which is not covered extensively in other literature review papers. All the bandits’ papers focus on maximizing the reward, while we rather use bandits as a tool to update the policy and emphasize potential outcome estimation.
>
> - While traditional introductions to causal inference emphasize linear models, modern machine learning techniques are well-suited for handling complex datasets. With the power of ML models, the key consideration becomes what kind of dataset we can practically gather and utilize, which heavily depends on the experimental setup. Therefore, we organized our approach into Active and Passive CI learning, aligning the CI algorithms with data collection procedures.

---

> ### Author Response · Authors · 2024-02-25
> **[2/3] Thank you for noticing that the intent of the paper is noble!**
>
> **Comment 1**: *“First, it only considers a single framework, that of potential outcomes, and disregards other existing frameworks without even a mention.”*
>
> RE: It is also unclear when reading the paper what are contributions from the authors, and what are existing frameworks and algorithms (we introduce, we estimate etc. are used freely at several places) … Overall, I do not find this paper to bring anything of value to the field or the community, especially in light of the plethora of existing introductory works in the literature [1-5].
>
> Our contribution is that we're looking at causal inference from two different angles: active and passive learning, which is different from how it is usually done. The methodologies referenced in [1-5] predominantly adopt a modeling-centric approach. While these methodologies serve a valuable purpose in theoretical education, they may not be optimal for individuals seeking practical applications in causal inference.
>
> We have structured our approach into active and passive sections, delineated by experimental setup. Many practitioners still prefer the active causal inference approach due to their discomfort with non-randomized datasets, whereas machine learning practitioners often prefer to formulate models for such datasets.
>
> Our conviction is that our paper furnishes a more accessible and straightforward introduction to causal inference for newcomers, particularly those with a strong inclination towards practical applications when juxtaposed with the works referenced in [1-5].
>
> **Comment 2**: *"In this paper [...] under such a causal graph." -> The introduction starts getting confusing here. Is it absolutely necessary to assume the availability of a causal graph to do CI? What does a causal graph allow for, which its absence does not allow? Until now it has not been said if the problem of interest was CI from interventional data, observational data, or a mix of both. The challenges of CI are not the same in each of these settings, and knowledge of the causal graph is not required in the purely interventional setting.*
>
> RE: Yes, we agree that a casual graph is not necessary to do causal inference. The list of casual graphs in Figure 1 is used to illustrate the relationship between confounder and other variables.
>
> We adjusted the writing in the paper (page 2) to clarify that certain methods don't require knowing the causal graph, such as RCT and difference-in-difference method, while other methods that use observational data require knowing the structure of a causal graph.
>
>
> **Comment 3**: *"We measure [...] treatment effects." -> What are average and individual treatment effects? This paper claims to be a starting point for people interested in but not familiar with causal inference, yet here it assumes right-away that the reader is familiar with non-trivial concepts.*
>
> RE: Changed to “We measure out how treatments affect outcomes by looking at both the average effect across groups and how each person's treatment affects them personally.”
>
> **Comment 4**: *"There exist other frameworks for CI, not mentioned here, such as the do-calculus framework of Judea Pearl in which these assumptions all boil down to a simpler condition, identifiability. If this paper only covers the potential outcomes framework of Donald Rubin, then it should be clearly stated. Unfamiliar readers (the target audience of the paper) might be lured to believe potential outcomes is the one and only framework for CI.*
>
> Yes, often most introductory papers do not mix between Rubin’s framework of potential outcomes and Judea Pearl’s framework of do-calculus.  This introductory paper diverges from conventional teaching methods in causal inference by combining both Rubin’s framework of potential outcomes and Judea Pearl’s framework of do-calculus. We adopt a holistic approach, drawing upon concepts from both paradigms to construct a foundational understanding of causal inference rooted in first principles.
>
> By incorporating aspects of both frameworks, we make the CI concept easier to understand potential outcomes alongside intervention probabilities. Our choice to introduce potential outcomes initially stems from its intuitive appeal, particularly when illustrating treatment effects through familiar examples from domains such as medicine or the sciences. As we formalize causal models, including intervention probabilities becomes natural to add to our toolbox alongside joint and conditional probability distributions.
>
> We acknowledge the need for clarity in delineating between these frameworks within the paper. To address this, we will explicitly state that we are integrating elements of both Rubin’s and Pearl’s approaches to elucidate causal inference concepts effectively.

---

> ### Author Response · Authors · 2024-02-25
> **[3/3] Thank you for noticing that the intent of the paper is noble!**
>
> **Comment 5**: *"Figure 1 -> these are graphs, not CI methods. What is the meaning of squares vs circles, and dark grey vs light grey nodes? What is the difference between the "deep latent models" and "deep causal models" graphs ?"*
>
> RE: Dark gray nodes correspond to observed variables while light gray nodes correspond to latent variables. A square node corresponds to a deterministic variable while a circle corresponds to stochastic variables. We made it clear in the caption between dark grey vs light grey nodes and square vs. circle nodes.
>
> Deep latent models correspond to deep neural network architecture with stochastic latent variables (e.g., [1,2] while deep causal models correspond to deep neural networks with deterministic hidden layers (e.g. [3,4]).
>
> [1] Christos Louizos and Uri Shalit and Joris Mooij and David Sontag and Richard Zemel and Max Welling. Causal Effect Inference with Deep Latent-Variable Models 2017
> [2] Rakesh, Vineeth and Guo, Ruocheng and Moraffah, Raha and Agarwal, Nitin and Liu, Huan Linked Causal Variational Autoencoder for Inferring Paired Spillover Effects 2018
> [3] Fredrik D. Johansson and  Uri Shalit and David Sontag. Learning representations for counterfactual inference 2016
> [4] Shalit, Uri and Johansson, Fredrik D. and Sontag, David Estimating Individual Treatment Effect: Generalization Bounds and Algorithms 2017
>
>
> **Comment 5**: *"We define the joint, conditional and intervention probabilities" -> Where does the intervention probability come from? This formula is valid only for specific graphs, which is not made clear from the text. Also, the do-calculus notation is used here, while only the potential outcome framework was referred to in the text earlier. This is inconsistent and confusing, especially for an introductory paper. Other information is missing, like p which is introduced without context or definition. Following what is said earlier in the introduction, which setting are we in here? Observational? Interventional? Passive? Active?"*
>
> We talked about international probability in the introduction (see page 1, “To infer causal effects we must measure intervention probabilities…”). We also mention in the introduction the scope of the problem (see page 2, “In this paper, we consider a problem setup in which we have a Covariate X, an action A, and an outcome Y.”
>
> Please see our comment above regarding Rubin’s and Pearl’s framework and our contribution statement at the very beginning.
>
> **Comment 6**: *"This is inconsistent with Algorithm 1, where potential outcome is . To this point, I am still unsure which setting is studied here. Are we in a bandit setting, where we are looking for an unconditional policy for ? Or are we in a contextual setting, where we want a policy for ?Why is Algorithm 1 sampling actions from a contextual policy, but then marginalizes out the covariate when computing the potential outcomes?*"
>
> RE: Please see the adjusted Algorithm 1.  Bandit algorithm updates policy function at each time step $pi_1$, $pi_2$, …, $pi_3$. The potential outcome is calculated using inverse probability weighting. During the exploitation phase, we IPW by p(a|x), and we divide by $p(a)$ during the exploitation phase (sampled from p(A|X=x)). For instance in Thompson sampling, then we would be always in the exploitation phase and divide by the p(a|x).
>
> **Comment 7**: *"there is a major difference -> I believe this is an overstatement. Most bandit algorithms keep track of the expected outcome of each action, in order to find the best policy. And, as stated in the introduction, CI models potential outcomes so that it can pick the action that maximizes the outcome."*
>
> The expected outcome and expected potential outcomes are different. We are borrowing policies from the bandit algorithm to do causal effect in online fashion.
>
> **Comment 8 **: *"We then estimate the average treatment effect (ATE) -> I find it very confusing that the authors frequently say "We do X", "We propose X", "We introduce X", while they also claim this paper to be an introduction to existing methods."*
>
> RE: Upon reviewing the Active CI section, we noted that phrases such as 'We propose' are not present. We respectfully disagree with the perception that the presented methods and algorithms imply original contributions. Our intention in this introductory paper is to provide a comprehensive overview of existing methods in the field of causal inference, including the application of bandit methods to estimate causal effects in dynamic environments. While we acknowledge that our discussion may imply novelty in the application or synthesis of these methods, we assure the reviewer and readers that our work is grounded in the existing literature. We welcome any specific references or suggestions the reviewer may have for further contextualizing our approach within the broader body of research in causal inference.
>
> All other misc comments are fixed!

---

### Author Response · Authors · 2024-02-25
**General response and summary of revision**

Thank you for your valuable feedback and insightful comments on the literature review. We appreciate positive feedback about the comprehensiveness and clarity of the papers. We have addressed your comments and edited the manuscript accordingly. We have marked all changes in pink (reviewer RLMu), green (reviewer 56z6), and blue (reviewer th1G).

We are happy to see that all the reviewers noticed our novel intention, which is to provide a unified framework for a broader audience. We have put thorough thoughts on the presentation and collections of existing works on causal inference.
Restating our contributions:

- Unlike other introduction to CI papers, we utilize both potential outcomes and do-calculus frameworks in our paper. We provide a comprehensive and unified perspective on causal inference, bridging the intuitive conceptualization of potential outcomes with the formal treatment of intervention probabilities.

- We look at causal inference from two different angles: active and passive learning, which is different from how it is usually done. While existing literature papers serve a valuable purpose in theoretical education, they may not be optimal for individuals seeking practical applications in causal inference.

- Our paper furnishes a more accessible and straightforward introduction to causal inference for newcomers, particularly those with a strong inclination towards practical applications when juxtaposed with the existing CI literature reviews. Indeed, the existing literature reviews require readers with some familiarity and background in machine learning and mathematics, whereas our paper aims to cater to a broader audience of machine learners at all levels of expertise.

---

### Decision · Action_Editor_wzR6 · 2024-03-30

**Recommendation:** Reject

**Comment:**

All the reviewers acknowledged that the authors indeed tried to provide a comprehensive introduction. However, unfortunately, they failed to do so. AE checked the background of the three reviewers: one does not know causal inference, and the other two are experts as they have published several papers in this area. All of them raised concerns about the inconsistencies, restricted scope, and confusing notations of the paper. After AE reading through the paper, AE feels the same. So, the paper is not friendly for beginners in causal inference as it does not develop new notations and scopes to better illustrate and explain the insights of causality.

AE recommends rejection.

**Audience:**

General audience who is knowledgeable in statistical machine learning.

**Claims And Evidence:**

The goal of the paper, as claimed by the authors, is to offer a comprehensive introduction to causal inference for machine learning researchers, practitioners, and students. Different from other popular causal inference books, this paper provides a new perspective: active and passive, where it further discusses causal identification, exchangeability, positivity, consistency, etc.

**Resubmission Of Major Revision:**

The authors may consider submitting a major revision at a later time.